# Machine learning-based prediction reveals kinase MAP4K4 regulates neutrophil differentiation through phosphorylating apoptosis-related proteins

Guihua Wang[1☉], Dan Zhang[1☉], Zhifeng He[1☉], Bin Mao[1], Xiao Hu[1], Li Chen[2], Qingxin Yang[2], Zhen Zhou[1], Yating Zhang[1], Kepan Linghu[1], Chao Tang[2], Zijie Xu[1], Defu Liu[1], Junwei Song[1], Huiying Wang[1], Yishan Lin[1], Ruihan Li[1], Jing-Wen Lin[2]*, Lu Chen[1]*

**1** Key Laboratory of Birth Defects and Related Diseases of Women and Children of MOE, Department of Laboratory Medicine, State Key Laboratory of Biotherapy, West China Second University Hospital, Sichuan University, Chengdu, Sichuan, China, **2** Biosafety Laboratory of West China Hospital, State Key Laboratory of Biotherapy, West China Hospital, Sichuan University, Chengdu, Sichuan, China

☉ These authors contributed equally to this work.
\* lin.jingwen@scu.edu.cn (JWL); luchen@scu.edu.cn (LC)

## Abstract

Neutrophils, an essential innate immune cell type with a short lifespan, rely on continuous replenishment from bone marrow (BM) precursors. Although it is established that neutrophils are derived from the granulocyte-macrophage progenitor (GMP), the molecular regulators involved in the differentiation process remain poorly understood. Here we developed a random forest-based machine-learning pipeline, NeuRGI (Neutrophil Regulatory Gene Identifier), which utilized Positive-Unlabeled Learning (PU-learning) and neural network-based *in silico* gene knockout to identify neutrophil regulators. We interrogated features including gene expression dynamics, physiological characteristics, pathological relatedness, and gene conservation for the model training. Our identified pipeline leads to identifying Mitogen-Activated Protein Kinase-4 (*MAP4K4*) as a novel neutrophil differentiation regulator. The loss of MAP4K4 in hematopoietic stem cells and progenitors in mice induced neutropenia and impeded the differentiation of neutrophils in the bone marrow. By modulating the phosphorylation level of proteins involved in cell apoptosis, such as STAT5A, MAP4K4 delicately regulates cell apoptosis during the process of neutrophil differentiation. Our work presents a novel regulatory mechanism in neutrophil differentiation and provides a robust prediction model that can be applied to other cellular differentiation processes.

**Data availability statement:** The codes used in this article were deposited in https://github.com/LuChenLab/NeuRGI. The control and Map4k4-cKO mice bone marrow scRNA-seq raw fastq files were deposited at NCBI under accession number GSE276482. For public data, we used the European Bioinformatics Institute (EBI) database of complete GWAS summary data that are publicly available from https://www.ebi.ac.uk/gwas/downloads/summary-statistics, the following study IDs were used to obtain the exposure data: neutrophil count (ebi-a-GCST004629). Mouse bone marrow bulk-RNA sequencing data are available in the GEO database with accession number GSE142216. Mouse bone marrow neutrophil single-cell sequencing data are available in the GEO database with accession number GSE243466. Human umbilical cord blood data are available in the European Genome-phenome Archive (https://ega-archive.org/) with accession number EGAD00001000745. PhastCons file phastCons100way.UCSC.hg38 was obtained from the UCSC Genome Browser (https://hgdownload.cse.ucsc.edu/goldenpath/hg38/phastCons100way/). The gene expression data used in OntoVAE model training for ImmuNexUT and atlas human lymphocytes are available in the National Bioscience Database Center (NBDC) Human Database and European Nucleotide Archive (ENA) under the accession code E-GEAD-397 and PRJEB5468, respectively.

**Funding:** This work was supported by the National Natural Science Foundation of China (82370233 to LC; 82341122 to J-wL; 92369116 to J-wL; and 82300133 to CT), Sichuan Science and Technology Program (2021YFS0027 to LC), and National Key Research and Development Program of China (2017YFA0106800 to LC). The funders had no role in study design, data collection and analysis, decision to publish, or preparation of the manuscript.

**Competing interests:** The authors have declared that no competing interests exist.

## Author summary

Hematopoiesis is a dynamic process driven by hematopoietic stem and progenitor cells (HSPCs) in the bone marrow. Neutrophils are the most abundant white blood cells in the human blood and play a key role in the innate immune response. Their main function is to act as the first line of defense against infections through phagocytosis, antimicrobial substance release, and extracellular trap formation. Neutrophil differentiation is known to be regulated by transcription factors like PU.1 and C/EBPα. However, the complexity of this process requires further study to identify key regulators. To identify critical drivers of neutrophil differentiation, we proposed a machine learning-based approach, Neutrophil Regulatory Gene Identifier (NeuRGI) based random forests and deep learning, which predicts genes involved in neutrophil differentiation based on neutrophil-specific features. This approach integrates dynamic expression profiles, disease relevance, gene conservation. Applying NeuRGI, we identified (Mitogen-Activated Protein Kinase Kinase Kinase Kinase 4 (*MAP4K4*) as a novel neutrophil differentiation regulator. Loss of MAP4K4 expression in HSPCs results in neutropenia, and MAP4K4 regulates cell apoptosis during neutrophil differentiation in bone marrow. This study provides new insights into neutrophil differentiation and offers a predictive model applicable to other differentiation processes.

## Introduction

Neutrophil differentiation is a critical process in the hematopoietic system, where stem cells in the bone marrow develop into mature neutrophils. Neutrophils are the most prevalent type of immune cells in human peripheral blood and act as the primary defender against both sterile injuries and microbial invasions. They deploy potent effector functions to neutralize external threats and play a crucial role in tissue remodeling [1]. Neutrophils have a short lifespan, with a half-life of approximately 19 hours in humans [2]. Consequently, there is a continual need to replenish neutrophils, and any disruption in their production or migration can lead to neutropenia, which can escalate into life-threatening complications [3].

Driver genes in hematopoiesis, including key regulators for cell differentiation, often govern critical signaling pathways that impact cell fate determination. Extensive research effort has been employed to identify key driver genes crucial for neutrophil differentiation. Transcription factors (TFs) such as PU.1, C/EBPα, and Gfi-1 have been identified as vital in neutrophil differentiation [4–6]. Ongoing research continues to unveil new layers of regulation during neutrophil differentiation. With technical advancement in single-cell RNA-sequencing (scRNA-seq), more insights have been gained into the complex and diverse regulatory networks during hematopoiesis [5,7,8].

Despite advances in this field, the complexity of neutrophil differentiation necessitates further investigation and identification of driver genes to explore this intricate biological process fully. Computational methods to identify key driver genes in complex biological processes are emerging. For example, approaches such as CellOracle [9] and SCENIC [10] were designed to screen for TFs, however other functional genes such as enzymes, membrane proteins, and RNA binding proteins (RBPs) were neglected.

Here we present a machine-learning-based approach to screen for key genes regulating neutrophil differentiation, NeuRGI (Neutrophil Regulatory Gene Identifier), which utilizes neutrophil cell-specific features to predict genes involved in regulating neutrophil differentiation. Our model provides prediction for all kinds of genes, including enzymes, membrane

proteins, and RBPs. We integrated 4 feature categories to train the model, including neutrophil-specific expression dynamics, physiological characteristics, pathological related-ness, and gene conservation. Our prediction model is based on random forest [11], which offers a balance in prediction accuracy and model interpretability when dealing with complex, high-dimensional data. Because the negative genes were difficult to define, we utilized the positive-unlabeled (PU) learning approach [12] for inferring potential negative genes from the unlabeled gene set. Moreover, we applied a deep-learning neural network to predict the effect of gene knockout [13], which provides important information that helps us to further explore the mechanism of the gene in regulating neutrophil differentiation. Using this analysis pipeline, we identified a novel neutrophil differentiation regulator, Mitogen-Activated Protein Kinase-4 (*MAP4K4*), a member of the Ste20p protein family, that has been reported to play pivotal roles in embryonic development [14,15], and cancer progression [16–18]. We found that the loss of MAP4K4 expression in HSPCs leads to neutropenia in mice and MAP4K4 regulates cell apoptosis during neutrophil differentiation.

## Results

### Development of the NeuRGI model

To identify genes important in neutrophil differentiation, we developed a supervised learn-ing algorithm NeuRGI (Neutrophil Regulatory Gene Identifier) (Fig 1A). We reasoned that a functional or driver gene should have all of the following four features: (1) differential or dynamic expressed during neutrophil differentiation. (2) associated with various physiological characteristics of neutrophils. (3) associated with related diseases. (4) conserved in multiple species.

We collected extensive datasets for feature extraction (**S1** and S2 **Tables**). The expression dynamics were calculated independently using three RNA-sequencing datasets, including bulk RNA-seq data of 8 FACS-sorted continuous hematopoietic populations in mouse bone marrow (BM) [19] and human umbilical cord blood [20], and scRNA-seq data of sorted CD11b$^+$Ly6G$^+$ mouse neutrophils [21]. Homologous genes between mice and humans were aligned using the BioMart database [22]. We calculated features such as gene expression range, cell specificity index Tau, and gene expression change rate extracted from the pseudotime trajectory inferred from scRNA-seq, to measure the differential and dynamic gene expres-sion fluctuations throughout neutrophil differentiation. For physiological characteristics, we categorized genes according to their associations with neutrophil physiological characteristics, such as neutrophil count and neutrophil measurement, based on Genome-Wide Association Studies (GWAS) information from the Open Targets Platform [23]. For pathological related-ness, we evaluated the Disease Specificity Index (DSI) from DisGeNET [24] and the number of neutrophil-related diseases (NofDisease) such as abnormal neutrophil count based on the MSigDB [25] databases. For conservation, we utilized PhastCons conservation scores from UCSC Genome Browser [26] which are derived from multiple alignment analyses of humans and 99 other vertebrate species.

The training model of NeuRGI is based on the random forest. To train the model, we encompassed a positive gene set (P set), containing 293 protein-coding genes from 19 Gene Ontology Biological Processes (GO-BP) pathways with the keyword 'neutrophil' or genes reported having neutrophil-related functions in the literature [8,27–30] (**S3 Table**). Next, we used the Positive-Unlabeled Learning (PU-learning) Spy algorithm [12] by adding 10% of the positive genes to an unlabeled set as 'spies'. We then trained a Naïve Bayes classifier with the new set of genes and assigned probabilities to each gene in the unlabeled set. Genes with probabilities below the 10$^{th}$ percentile were considered 'negative' (N set, see **Methods**). The

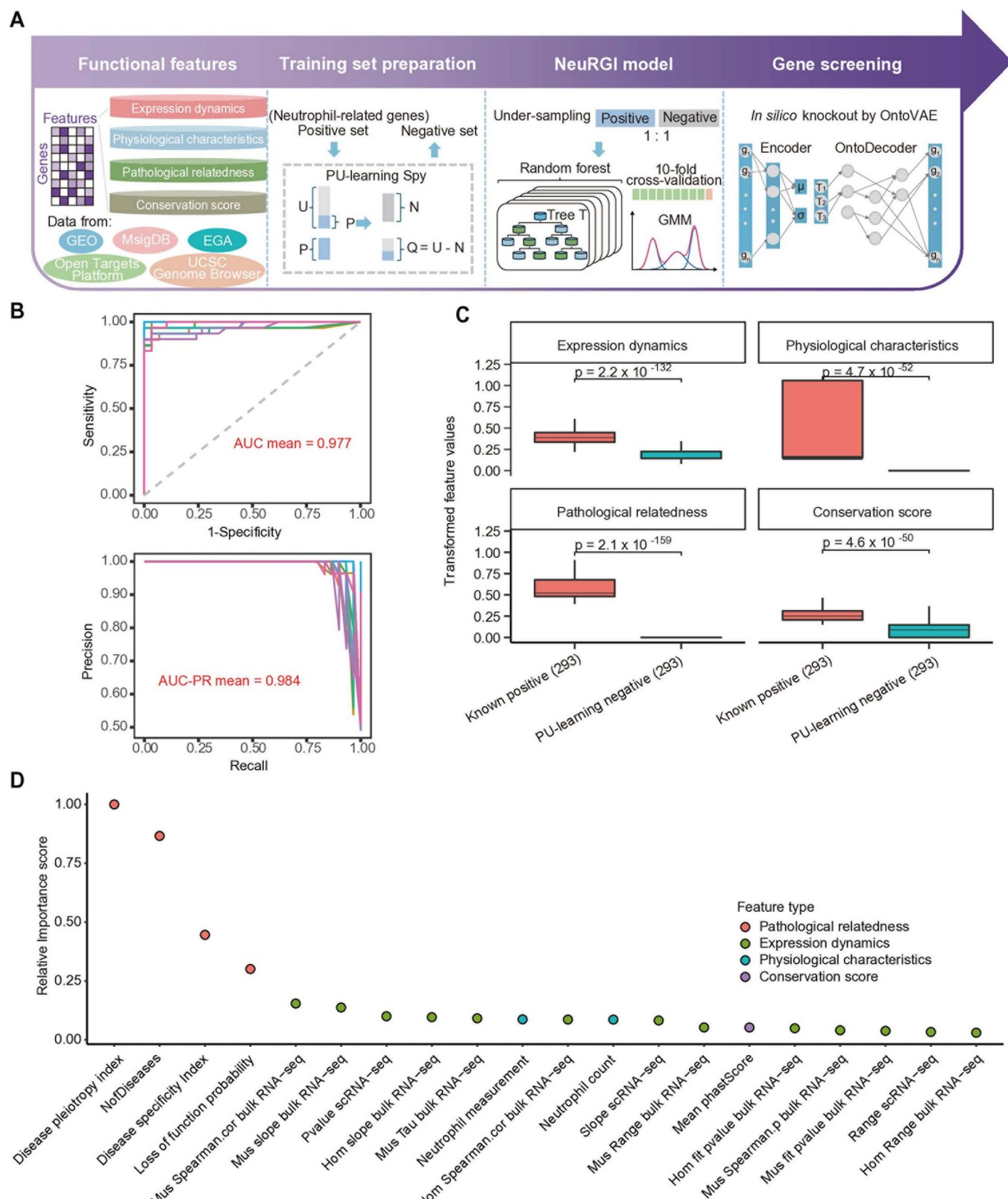

**Fig 1. Development of a machine learning method for predicting neutrophil-specific functional genes.** (A) NeuRGI model training workflow involved: 1) extracting gene features from various databases. 2) using genes of neutrophil-related genes as positives and PU-learning as negatives. 3) balancing the

training set with under-sampling and training the NeuRGI random forest model with 10-fold cross-validation, then employing a Gaussian Mixture Model (GMM) with NeuRGI scores to identify potential positives. 4) using OntoVAE for *in silico* knockout of GMM-classified genes to find key regulatory factors for guiding follow-up experiments. (B) AUC and PR curves and the mean AUC value for the NeuRGI model trained from 10-fold cross-validation. The data is split into 10 parts, with 9 parts for training and 1 for testing, repeated 10 times with each part used once for testing. (C) The boxplot illustrates the transformed feature values for 293 known positive genes and 293 PU-learning negative genes in four feature groups. The p value was calculated using the Student's t-test. (D) Distribution of the relative importance score calculated by average mean decrease Gini for features used in NeuRGI model (color-coded by category).

under-sampling method [31] was used to obtain a balanced training data set, consisting of 293 positive and 293 negative genes (S3 Table). The GO-BP enrichment analysis of the 'N' set indicated that the genes were not related to neutrophil differentiation (S1A Fig).

We performed a random forest model fitting on this training set using 10-fold cross-validation. The importance of all features was evaluated based on their relative importance scores of the Gini coefficient (S1B Fig), and feature ablation studies were conducted. We found that the model, after removing 7 bottom-ranked features (Neutrophil to lymphocyte ratio, Sum of neutrophil and eosinophil counts, Neutrophil percentage of leukocytes, Neutrophil percentage of granulocytes, Abnormal neutrophil count, Hom Tau bulk RNA-seq, Cor scRNA-seq and Hom Spearman.p bulk RNA-seq), exhibited the best overall performance (AUC = 0.977, AUC-PR = 0.984, ACC = 0.962, MCC = 0.927, F1 Score = 0.964, S1C Fig and S4 Table). Therefore, this refined model was established as the NeuRGI model (Fig 1B). We further compared the normalized feature scores between the positive and negative sets regarding gene expression, physiological characteristics, pathological relatedness, and gene conservation. The relative scores in the positive genes were significantly higher than those in the negative genes in all four feature categories (Fig 1C). According to the relative importance score of the features, we found that the most predictive feature is pathological relatedness, constituting 68% of the total importance, followed by expression dynamics (26%) (Fig 1D).

## Benchmarking NeuRGI for inferring functional genes

We applied NeuRGI to 19,288 protein-coding genes that were not used in model training, and assigned prediction scores for each gene, ranging from 0 to 1, with 1 as most likely to be 'neutrophil regulator' (S5 Table). We observed that the distribution of the NeuRGI score exhibited a bimodal distribution, with two peaks around 0.02 and 0.97 identified using R package quantmod (v0.4.26) [32], respectively (Fig 2A). We thus fitted the NeuRGI scores of all predicted genes with a three-component Gaussian Mixture Model (GMM) and categorized the genes into 3 classes: 'functional' in neutrophil differentiation, with NeuRGI scores higher than 0.96; 'non-functional', with scores lower than 0.02; and 'uncertain', with the scores between 0.02 to 0.96. (Fig 2A). In total, we identified 4,786 genes likely to be functional in neutrophil differentiation, and 4,734 genes unlikely to be involved in neutrophil differentiation (S1D Fig).

We next compared NeuRGI with CellOracle [9] and SCENIC [10], two state-of-the-art tools for inferring gene regulatory networks (GRN), based on perturbation scores and regulon activity, respectively. Using scRNA-seq data of 2,803 BM neutrophils [21], CellOracle and SCENIC identified 70 and 36 TFs, respectively (S6 Table). The majority of these TFs (68.6% for CellOracle and 75.0% for SCENIC) were classified as functional by NeuRGI (S1E Fig). Additionally, all of the top 20 TFs ranked by CellOracle's perturbation score and SCENIC's regulon activity overlapped with NeuRGI's predicted functional set (Fig 2B), demonstrating accuracy of NeuRGI. Notably, 22 and 9 TFs that were predicted to be important by CellOracle or SCENIC were predicted as 'uncertain' by NeuRGI. We further explore the biological distinctions between the 'uncertain' and the 'functional' TFs called by NeuRGI and one of the other methods. When comparing four feature values across the two groups, we observed

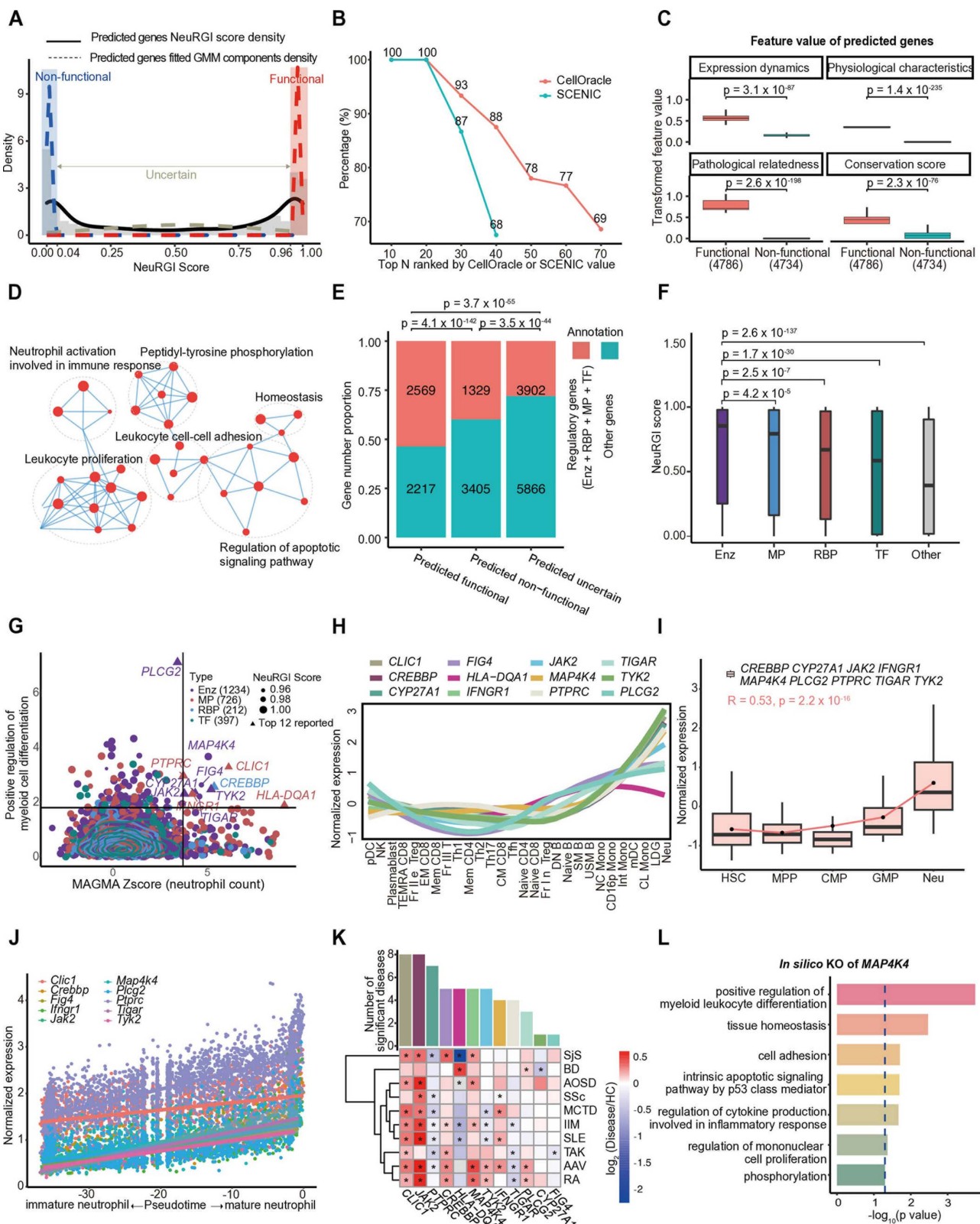

**Fig 2. Machine learning reveals *MAP4K4* as a novel regulator of hematopoietic neutrophil differentiation.** (A) Three-component Gaussian Mixture Models (GMM) were applied to the NeuRGI scores across all 19,288 predicted genes. The solid black line represents the overall score distribution, dashed lines indicate the three probability density functions (PDFs). Intersections of the three PDFs define NeuRGI thresholds for classifying

genes into function (red shadow), non-function (blue shadow), and uncertain (middle white) categories. (B) The line plot depicts the overlap of functional transcription factors (TFs) predicted by CellOracle and NeuRGI (red line), and by SCENIC and NeuRGI (blue line). The x-axis represents the top **N** TFs ranked by CellOracle's perturbation score or SCENIC regulon activity. The y-axis shows the percentage of overlap between the predicted functional TFs. (C) The boxplot illustrates the transformed feature values for 4,786 functional genes and 4,734 non-functional genes in four feature groups. The p value was calculated using the Student's t-test. (D) The GO-BP term network illustrates the five main clusters enriched from 4,786 predictive functional genes (see Methods). Each dot represents a GO term and the dot size indicates the enrichment score. The dashed oval indicates GO terms with similar functions. (E) Proportion of regulatory (red) and other (blue) genes across three gene sets. Regulatory genes include enzyme (Enz), membrane protein (MP), RNA binding protein (RBP), and transcription factor (TF). The p value was calculated by the proportion test. Numbers within bars represent the gene counts. (F) Boxplot of NeuRGI scores, color-coded by gene type: Enzyme (Enz), membrane proteins (MP), RNA binding proteins (RBP), Transcription factors (TF), and others. The p value was calculated using the Student's t-test. (G) The scatter plot shows the impact of *in silico* knockout of 2,569 predictive functional regulatory genes on the "positive regulation of myeloid cell differentiation" pathway and MAGMA Zscore (neutrophil count). The y-axis represents the $-\log_{10}$ (p value) on the pathway after gene *in silico* knockout (higher values indicate greater impact), and the x-axis represents the gene's effect on the 'neutrophil count' trait (higher Zscores indicate greater impact). Different colors represent different categories of genes. Dot size indicates the NeuRGI score, and contour lines show point density. A cutoff (y = 1.8 and x = 3.6) was set based on the contour lines, dividing the scatter plot into four regions. (H) Expression of top 12 genes in different immune cells from ImmuNexUT, including Naïve CD4 T cells (Naïve CD4), Memory CD4 T cells (Mem CD4), T helper 1 cells (Th1), T helper 2 cells (Th2), T helper 17 cells (Th17), T follicular helper cells (Tfh), Fraction II effector regulatory T cells (Fr. II eTreg), Fraction I naïve regulatory T cells (Fr. I nTreg), Fraction III non-regulatory T cells (Fr. **III** T), Naïve CD8 T cells (Naïve CD8), CD8+ T effector memory CD45RA+ cells (TEMRA CD8), Effector Memory CD8 T cells (EM CD8), Central Memory CD8 T cells (CM CD8), Naïve B cells (Naïve B), Unswitched memory B cells (USM B), Switched memory B cells (SM B), Double Negative B cells (DN B), Plasmablasts (Plasmablast), Natural Killer cells (NK), CD16 positive monocytes (CD16p Mono), Non-classical monocytes (NC Mono), Intermediate monocytes (Int Mono), Classical monocytes (CL Mono), Myeloid dendritic cells (mDC), Plasmacytoid dendritic cells (pDC), neutrophils (Neu), Low-Density Granulocytes (LDG). (I) Expression of 9 genes in neutrophil differentiation of human UCB. These 9 of the top 12 genes dynamically upregulated, including *CREBBP, CYP27A1, JAK2, IFNGR1, MAP4K4, PLCG2, PTPRC, TIGAR,* and *TYK2*. We set 'time cut' for cells at different differentiation stages, with HSC set as 1 and Neu as 5, and performed linear regression fitting for the expression of all 9 genes. R represents the Pearson correlation coefficient (R), and the p value (p) was calculated by the Student's t-test. (J) Expression of top 10 genes in single-cell pseudotime analysis of 2,803 neutrophils in mouse bone marrow. All 12 genes except *Cyp27a1* and *HLA-DQA1* upregulated during neutrophil maturation. (K) The heatmap displays the $\log_2$ (fold change) in the expression of the top 12 genes in neutrophils from patients with 10 immune-related diseases compared to those from healthy individuals, including ANCA-associated vasculitis (AAV), Takayasu arteritis (TAK), Adult-onset Still's disease (AOSD), Behçet's disease (BD), Rheumatoid arthritis (RA), Systemic sclerosis (SSc), Idiopathic inflammatory myopathy (Myo), Sjögren's syndrome (SjS), Mixed connective tissue disease (MCTD), Systemic lupus erythematosus (SLE). The p value was calculated using the Student's t-test, and * represents statistical significance (p < 0.05). The histogram represents the number of significant diseases for each gene. (L) Bar plot displaying significantly affected pathways after OntoVAE *in silico* knockout of *MAP4K4* in neutrophils. The blue dashed line represents the significance threshold (p = 0.05).

that consistently functional TFs showed significantly higher pathological relatedness than uncertain TFs (p < 0.012, S1F Fig), further showing the accuracy of NeuRGI in predicting functional TFs.

We evaluated the predicted functional and non-functional sets regarding the four feature categories scores (gene expression dynamics, physiological characteristics, pathological relatedness and gene conservation), experimental validation, GO-BP pathway, and the enrichment of regulatory genes (defined as TFs, RBPs, enzymes, or membrane proteins).

First, the feature values of the predicted 'functional' set were significantly higher than those of the predicted 'non-functional' set (Fig 2C), consistent with the observation in the model training. Second, 5 genes with NeuRGI scores higher than 0.995 were reported to play important roles in neutrophil differentiation or function. The 5 genes include 3 membrane proteins *FAS*, *IL17RA,* and *CD226*, 1 TF *STAT6, and* 1 enzyme *GRK6. FAS* and *STAT6* regulate neutrophil apoptosis [33,34]; *IL-17RA* plays a crucial role in neutrophil recruitment induced by IL-17 [35]. *CD226* (*DNAM-1*) modulates neutrophil infiltration and inflammation [36]; *GRK6* plays an important negative regulatory role in neutrophil-mediated inflammatory responses [37]. Finally, we conducted GO-BP analysis for all 4,786 functional genes and categorized the top 100 GO terms based on their similarity. The terms were clustered into 6 categories (S1G Fig), among which 'Neutrophil activation involved in immune', 'leukocyte proliferation', 'leukocyte cell-cell adhesion', 'regulation of apoptotic signaling pathway', 'Peptidyl-tyrosine phosphorylation', and 'homeostasis' were highly connected (Fig 2D). Notably, among the 4,786 genes identified as likely functional in neutrophil differentiation, 53.7% (2,569) genes were predicted to encode

regulatory proteins, including enzymes (1,234), membrane proteins (726), transcription factors (TFs, 397), and RNA-binding proteins (RBPs, 212). This proportion is significantly higher than that in the negative gene set (28.1%, Fig 2E, proportion test, p = 4.1 x 10$^{-142}$), suggesting the predicted functional set was enriched with regulatory genes. Interestingly, the NeuRGI score of enzymes was significantly higher than the other protein types (Fig 2F).

Taken together, NeuRGI demonstrated high accuracy in predicting functional regulators particularly those with high pathological relevance. Unlike CellOracle and SCENIC, NeuRGI extends its predictions beyond TFs, capable of identifying a broader range of functional genes. And NeuRGI-predicted functional genes are enriched for regulatory roles, exhibit higher feature scores, and are closely linked to key immune processes. These findings highlight NeuRGI as a comprehensive and robust tool for functional gene inference.

## Machine-learning strategy identified *MAP4K4* as a novel regulator of hematopoietic neutrophil differentiation

We performed *in silico* knockout in neutrophils for the 2,569 genes using Ontology-guided Variational Autoencoder (OntoVAE) [13], a deep learning predictive model trained on the 2,040 RNA-seq datasets of immune cells from ImmuNexUT [38] and Atlas Human Lymphocytes [39], with stable loss after 300 epochs (S1H Fig, see S1 Text). OntoVAE is a modified VAE architecture that can incorporate Gene Ontology and provide pathway activities in its latent space and decoder. We retrieved the activities of all pathways and found the 'neutrophil activation' term was especially associated with neutrophils, and 'regulation of mononuclear cell proliferation' was associated with monocytes, as expected (S1I Fig). *In silico* knockout was performed by setting the input of a gene as 0, then using paired Wilcoxon tests to identify the affected terms (pre-knockout versus post-knockout). To test the validity, we performed *in silico* knockouts of the *ELANE* [40] and *SYK* [41], two genes have been reported to play important roles in neutrophils and B cells, respectively. We found that pathways related to neutrophils and B cells were significantly affected in consistent with previous studies [42–45] (S2A Fig), suggesting that applying OntoVAE on RNA-seq data of immune cells can predict the pathways altered upon gene knockouts in neutrophils.

Next, we assessed the impact of the genes on the 'positive regulation of myeloid cell differentiation' by OntoVAE *in silico* knockout and calculated gene-level association Z score of 'neutrophil count' based on GWAS using MAGMA [46] (Fig 2G and S5 Table). We then identified 11 genes with top scores of both features and 1 gene with the highest scores of each feature (Fig 2G). All 12 genes have the highest expression in neutrophils in healthy individuals (Fig 2H), among which 9 genes showed increased expression along the neutrophil differentiation (from HSC to neutrophils) in humans (Fig 2I). In addition, the pseudotime trajectory analysis of scRNA-seq data using monocle2 [47] indicated that expression of the 10 genes increased from immature neutrophils (imNeu) to mature neutrophils (mNeu) in mice [21] (Fig 2J). The 12 genes were significantly differential expressed between healthy individuals and patients with immune-mediated diseases (Fig 2K). For example, *CLIC1* and *JAK2* were significantly differential expressed in 8 types of immune cell-related diseases, followed by *PTPRC* (7), *CREBBP* (5), *HLA-DQA1*(5), and *MAP4K4* (5).

The 12 genes include 7 enzymes (*PLCG2, JAK2, CYP27A1, TIGAR, MAP4K4, TYK2*, FIG4), 4 membrane proteins (*PTPRC, HLA-DQA1, IFNGR1* and *CLIC1*), 1 RNA binding protein (*CREBBP*). Nine genes have been reported to play functional roles in the neutrophils. Specifically, *JAK2* influences neutrophil production by regulating progenitor cell proliferation [48]. *PTPRC (CD45)* regulates neutrophil activation and migration post-myocardial infarction [49]. CLIC1 affects oxidative responses in neutrophils [50]. *IFNGR1* enhances immune resistance

to bacterial infections [51]. Additionally, *TYK2* modulates neutrophil function in acute inflammation [52]. The *CREBBP* is important for immune responses via oxidative burst regulation [53]. *PLCG2* affects neutrophil chemotaxis and inflammation in arthritis [54], *CYP27A1* may serve as a biomarker for intervertebral disc degeneration [55], and *HLA-DQA1* influences neutrophil-mediated immune processes [56]. The other 3 genes, *MAP4K4*, FIG4, and *TIGAR* have not been reported to be involved in neutrophil differentiation or functions, among which *MAP4K4* has the highest NeuRGI score (0.9887, S2B Fig). *MAP4K4* has the highest expression in neutrophils and highest neutrophil specificity (Tau = 0.75, S2C Fig) and the highest impact on the 'positive regulation of myeloid leukocyte differentiation' pathway in knockout simulation ($p = 1.79 \times 10^{-4}$, Figs 2L and S2D). The expression is increased along the neutrophil differentiation (S2E Fig, *R* = 0.73, Pearson correlation) (S7 Table).

## MAP4K4 is required for neutrophil differentiation

To investigate the role of MAP4K4 in neutrophil differentiation, HL-60 human leukemia cells were used in the *in vitro* studies. HL-60 can be differentiated into neutrophil-like cells characterized by CD11b expression upon stimulation with all-trans-retinoic acid (ATRA) [57] (Fig 3A). We found that MAP4K4 specific inhibitor, PF-06260933, significantly impeded the differentiation of HL-60 cells into CD11b+ cells in a dose-dependent fashion (Fig 3B). The percentage of CD11b+ cells was halved at the concentration of 5 μM and fell to ¼ of the control at 20 μM (Fig 3B), suggesting that MAP4K4 may be involved in neutrophil differentiation.

To confirm MAP4K4 plays a role in neutrophil differentiation *in vivo*, we generated transgenic mice whose hematopoietic stem cells and progenitors (HSPCs) lacking the expression of MAP4K4 (*Map4k4*-cKO, cKO) by crossing *Map4k4*<sup>fl/fl</sup> mice with interferon-inducible Mx1-Cre mice (S3A Fig). RT-PCR analysis confirmed the loss of *Map4k4* transcript in HSPCs after poly:IC treatment (S3B Fig). The total cell number in bone marrow was not changed in the cKO mice (S3C Fig). Furthermore, the percentages and the cell numbers of hematopoietic stem cells (HSCs) and progenitors were also not affected compared to controls (Ctrl) (*Map4k4*<sup>−/−</sup> MX1-Cre+) (S3D and S3E Fig). However, the colony formation analysis showed significantly reduced colony numbers derived from granulocyte-macrophage progenitor cells (CFU-GM, CFU-G, CFU-M) (Fig 3C), while primitive erythroid progenitor cells (BFU-E) remained unaffected (S3F Fig). In the investigation conducted using flow cytometry analysis, it was observed that there was a 15% reduction in the neutrophil count in cKO mice (Fig 3D), while the levels of eosinophils and monocytes were not affected (S3G and S3H Fig), indicating that the loss of MAP4K4 in HSPCs inhibits neutrophil differentiation.

cKO mice displayed a reduction of white blood cell count in the peripheral blood (Fig 3E), specifically monocyte and granulocyte counts were reduced by 68% and 50%, respectively (Fig 3F). In line with the *in-silico* KO prediction, lymphocyte, RBC, and platelet counts were not affected (S3I–S3K Fig). Using flow cytometry analysis, we confirmed that the number of neutrophils in peripheral blood was decreased by about 45% in cKO mice eosinophils were reduced by 50%, and monocytes were reduced by 45% (Fig 3G–3I). Similarly, the total splenocyte number also decreased in cKO mice (Fig 3J). Specifically, the numbers of neutrophils were reduced by 40%, and both eosinophils and monocytes were all reduced by 70% in the spleen (Fig 3K–3N). In contrast, the numbers of dendritic cells (DCs) and red pulp macrophages remained unaffected (S3l–S3M Fig).

Collectively, our machine-learning model identified a protein kinase, MAP4K4, which is required for neutrophil differentiation. Subsequent experiments confirmed that the deficiency of MAP4K4 in HSPCs results in a significant reduction in the populations of neutrophils and monocytes.

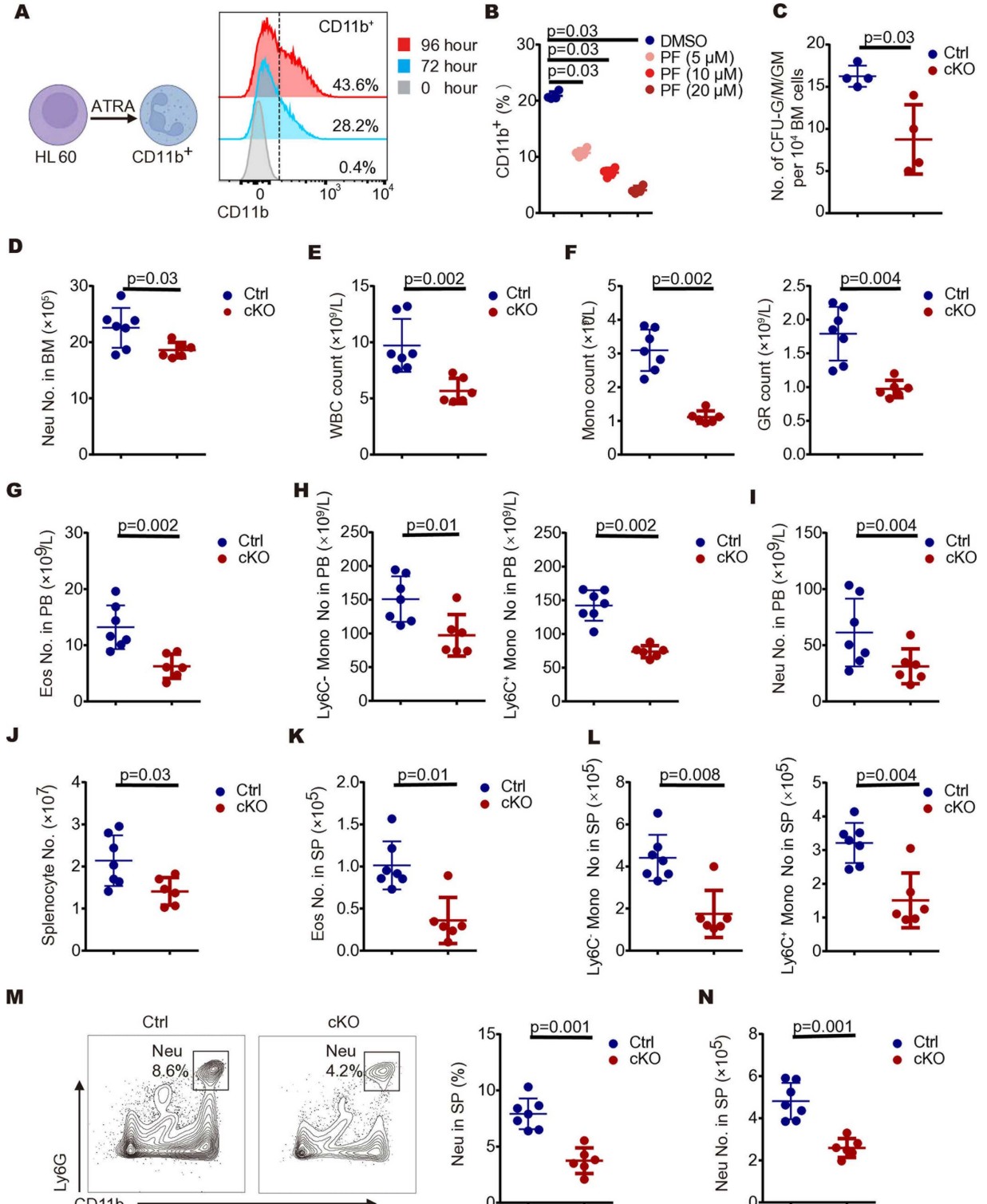

**Fig 3. Loss of MAP4K4 expression causes a decrease in the number of neutrophils.** (A) Scheme of all-trans-retinoic acid (ATRA)-induced neutrophil differentiation of the hematopoietic HL-60 cell models (left); flow cytometry analysis of the neutrophil marker CD11b has validated that ATAR can induce differentiation of HL-60 cells into neutrophils. This is demonstrated by histogram plots showing variations in mean fluorescence intensity (MFI) across different conditions: cells untreated with ATAR, cells treated with ATAR for 72 hours, and cells treated with ATAR for 96 hours (right). (B) Percentage of CD11b+ cells in HL-60 cells were treated with 0-20 μM MAP4K4 inhibitor (PF-06260933, PF) for 48 hours, followed by a subsequent 96-hour treatment with both PF and all-trans-retinoic acid (ATRA). (C) The number

of granulocyte-macrophage progenitor cells (CFU-GM, CFU-G, CFU-M) colonies formed by 25,000 whole bone marrow cells from control (Ctrl) or *Map4k4*-cKO (cKO) mice; BM, bone marrow (n=4; mean ± SD). (D) Numbers of neutrophils in the bone marrow of control (Ctrl) or *Map4k4*-cKO (cKO) mice; BM, bone marrow; Neu, neutrophil. (Ctrl n=7, cKO n=6; mean ± SD). (E) PB WBC numbers of control (Ctrl) or *Map4k4*-cKO (cKO) mice (Ctrl n=7, cKO n=6; mean ± SD). PB, peripheral blood; WBC, white blood cell; (Ctrl n=7, cKO n=6; mean ± SD). (F) PB Mon, and GR numbers of control (Ctrl) or *Map4k4*-cKO (cKO) mice; PB, peripheral blood; Mon, monocyte; GR, granulocyte; (Ctrl n=7, cKO n=6; mean ± SD). (G) Numbers of eosinophils in PB of control (Ctrl) or *Map4k4*-cKO (cKO) mice; PB, peripheral blood; Eso, eosinophils; (Ctrl n=7, cKO n=6; mean ± SD). (H) Numbers of monocytes in PB of control (Ctrl) or *Map4k4*-cKO (cKO) mice; PB, peripheral blood; Mon, monocytes; (Ctrl n=7, cKO n=6; mean ± SD). (I) Numbers of neutrophils in PB of control (Ctrl) or *Map4k4*-cKO (cKO) mice; PB, peripheral blood; Neu, neutrophil; (Ctrl n=7, cKO n=6; mean ± SD). (J) Spleen cell numbers control (Ctrl) or *Map4k4*-cKO (cKO) mice (Ctrl n=7, cKO n=6; mean ± SD). (K) Numbers of eosinophils in the spleen of control (Ctrl) or *Map4k4*-cKO (cKO) mice; SP, spleen; Eso, eosinophils; (Ctrl n=7, cKO n=6; mean ± SD). (L) Numbers of monocytes in the spleen of control (Ctrl) or *Map4k4*-cKO (cKO) mice; SP, spleen; Mon, monocytes; (Ctrl n=7, cKO n=6; mean ± SD). (M) Representative FACS analysis of spleen CD11b+ Ly6G+ neutrophils (left); percentage of neutrophils in the spleen of control (Ctrl) or *Map4k4*-cKO (cKO) mice (right); SP, spleen; Neu, neutrophil; (Ctrl n=7, cKO n=6; mean ± SD). (N) Numbers of neutrophils in the spleen of control (Ctrl) or *Map4k4*-cKO (cKO) mice; SP, spleen; Neu, neutrophil; (Ctrl n=7, cKO n=6; mean ± SD). Mann-Whitney U test.

## MAP4K4 regulates neutrophil differentiation in bone marrow

To further delineate the role of MAP4K4 in neutrophil differentiation, we performed scRNA-seq analysis on BM cells from cKO mice. Given the low abundance of HSPCs, we performed a sample enrichment for c-Kit+ HSPCs and mixed them with total BM cells at a 2:3 ratio for scRNA-seq (Fig 4A). Ctrl and cKO datasets were merged and anchor genes [58] were used to remove the batch effect (S4A Fig). After quality control, 9,051 cells from wildtype and 5,674 cells from cKO mice were included for further analysis. After removing mitochondrial and ribosomal genes, a total of 54,838 genes were detected with 11,261 reads (S4B Fig), an average of 2,865 genes per cell (S4C Fig), highlighting the high quality of our dataset (S8 Table). We next used Seurat [58] to process the 14,725 cells for normalization, batch effect correction, and cell-type clustering. The cells were visualized by Uniform Manifold Approximation and Projection (UMAP) and clustered into 16 distinct cell types based on the expression of canonical cell markers [7] (S4D Fig and S9 Table), including HSC and multipotent progenitors (HSC/MPP), granulocyte monocyte progenitors (GMP), neutrophil progenitors (ProNeu), neutrophil precursors (PreNeu), immature neutrophils (imNeu), mature neutrophils (mNeu), megakaryocytic-erythroid progenitors (MEP), erythrocyte progenitors (ProEry), monocyte progenitors (ProMon), B progenitors (ProB), B cells, natural killer cells (NK), natural killer T cells (NKT), eosinophils, macrophages and conventional dendritic cells (cDC) (Fig 4B and S4E Fig).

To model the neutrophil differentiation process, we applied Monocle2 [47] to 11,031 cells including HSC/MPP, GMP, and neutrophils of different maturation stages, and identified a continuously increasing pseudotime from HSC/MPP to mNeu, consistent with the neutrophil differentiation process [7] (S4F–G Fig). During this neutrophil differentiation, we found that the fractions of proNeu and preNeu in the cell division stage of neutrophil differentiation were increased in cKO mice. However, imNeu was decreased (Fig 4C). Aiming to validate these results from scRNA-seq, we analyzed the maturation stages of neutrophils in BM by flow cytometry. In line with the scRNA-seq data, we found a significant increase in promyelocytes (proNeu) [7], myelocytes (mixture of GMP and proNeu) [7], and metamyelocytes (preNeu) [7] in cKO mice BM (Fig 4D) and a reduction of band cells and segmented neutrophils (imNeu/mNeu) [7] in cKO mice BM (Fig 4D). To investigate the whether MAP4K4 affects the function of neutrophils, we assessed the reactive oxygen species (ROS) production in neutrophils and the precursors in BM. No significant difference in ROS levels were found among all of these populations in the bone marrow of MAP4K4-deficient mice compared to the control mice (Fig 4E). Similarly, neutrophils isolated from the spleen and peripheral blood

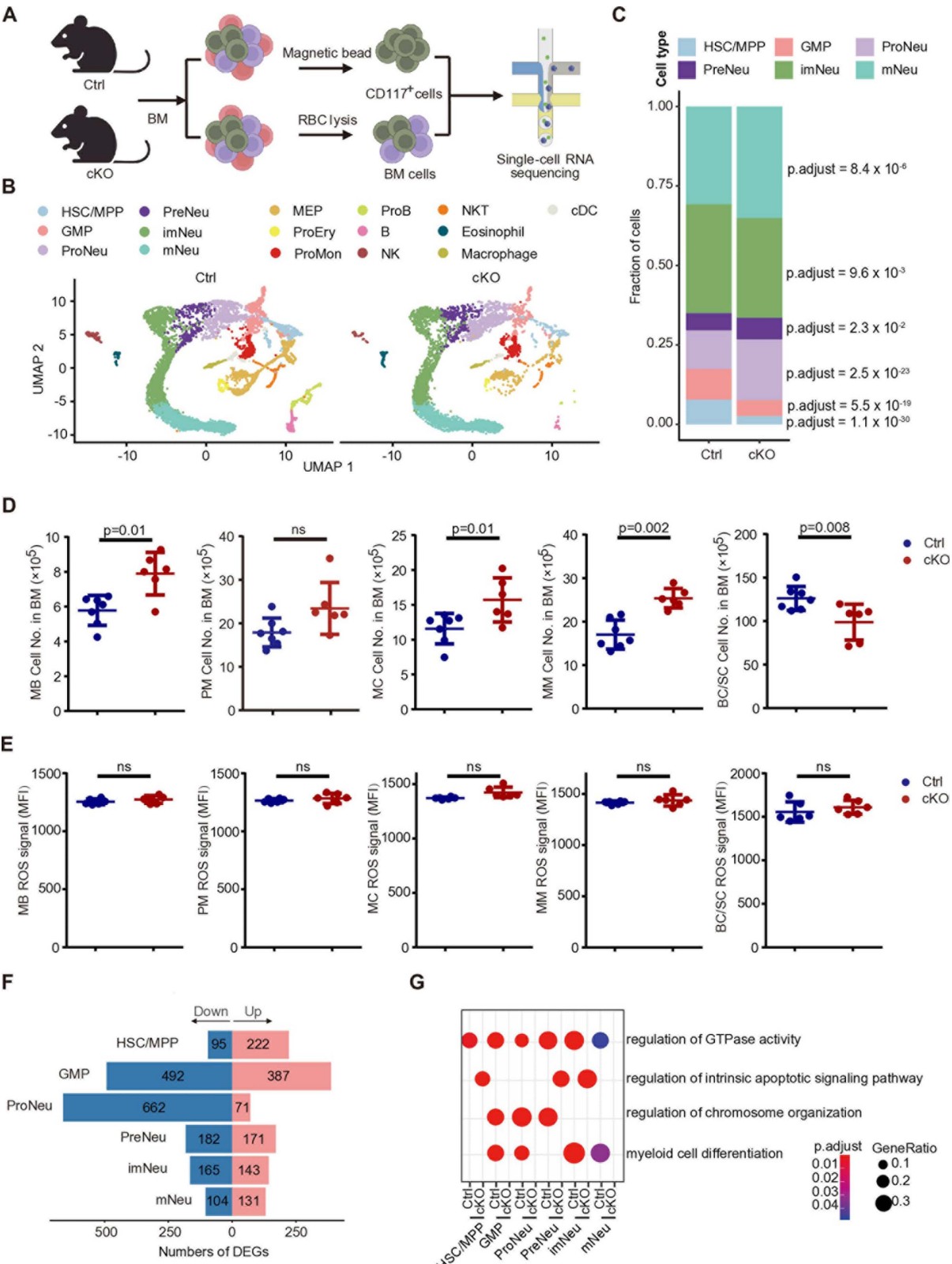

**Fig 4. Loss of MAP4K4 expression impairs the differentiation process of bone marrow neutrophils.** (A) Overview of study design, created using Biorender. (B) UMAP Plots for single-cell gene expression pooled across bone marrow samples, clusters visualized and labeled

by cell type. The plot is split by control (Ctrl) vs *Map4k4*-cKO (cKO). (C) Proportions of the six neutrophil clusters in control (Ctrl) and *Map4k4*-cKO (cKO) samples. The p value was calculated by the adjusted proportion test. (D) Numbers of neutrophil progenitor cells in the bone marrow of control (Ctrl) or *Map4k4*-cKO (cKO) mice; c-Kit$^{hi}$Ly6G$^{neg}$ (myeloblasts, MB); c-Kit$^{int}$Ly6G$^{neg}$ (promyelocytes, PM); c-Kit$^{neg}$Ly6G$^{low}$ (myelocytes, MC); c-Kit$^{neg}$Ly6G$^{int}$ (metamyelocytes, MM); and c-Kit$^{neg}$Ly6G$^{hi}$ (band cells and segmented neutrophils, BC/SC); BM, bone marrow (Ctrl n=7, cKO n=6; mean ± SD). Mann-Whitney U test. (E) ROS generation of neutrophil progenitor cells in BM of control (Ctrl) or *Map4k4*-cKO (cKO) mice; MFI, mean fluorescent intensity; ROS, reactive oxygen species; MB, myeloblasts; PM, promyelocytes; MC, myelocytes; MM, metamyelocytes; BC/SC, band cells and segmented neutrophils; (n=6; mean ± SD). Mann-Whitney U test. (F) The number of DEGs in control (Ctrl) vs *Map4k4*-cKO (cKO) of six neutrophil subpopulations. (G) GO-BP analysis of cluster-based DEGs between control (Ctrl) and *Map4k4*-cKO (cKO) HSC to neutrophils. Selected GO terms with Benjamini-Hochberg-corrected p values < 0.05 (one-sided Fisher's exact test) are shown. The dot size represented the number of genes. The color scale represented the adjusted p value. Fig 4A was created using Biorender.

of MAP4K4-deficient mice exhibited no significant alterations in ROS production (S4H Fig). Furthermore, the phagocytosis of *E. coli* by neutrophils from both the spleen and peripheral blood also showed no significant differences in the MAP4K4-deficient mice (S4I Fig). These results suggest that the deletion of MAP4K4 primarily affects the differentiation of neutrophils, with no significant impact on their function, such as ROS production and phagocytic activity.

## MAP4K4 regulates cell apoptosis during neutrophil differentiation process

To investigate the downstream regulatory network of MAP4K4, we performed differential expression gene (DEG) analysis for neutrophil subpopulations. ProNeu and GMP were the most affected populations with the largest number of DEGs (Fig 4F and S10 Table). Next, we compared the differences in pathways in Ctrl and cKO mice. Notably, the cKO up-regulated DEGs in HSC/MPP, PreNeu, and imNeu were specifically enriched with 'regulation of intrinsic apoptotic signaling pathway', while those up-regulated genes in GMP, ProNeu, imNeu, and mNeu of Ctrl mice were uniquely enriched with 'myeloid cell differentiation' (Fig 4G), suggesting MAP4K4 may potentially affect apoptotic signaling pathway.

Next, we further performed consensus co-expression network analysis using hdWGCNA [59] in scRNA-seq. Five co-expression modules (M) were identified (Figs 5A and S5A–S5E). Notably, the hub genes of M1 are mainly expressed in HSC/MPP, while M2 for GMP, ProNeu, and PreNeu, M3 in PreNeu, M4 in imNeu, and M5 in mNeu. For instance, classical marker genes of mNeu including *Retnlg* and *Mmp8* were highly expressed in M5 (Fig 5B). These results showed that these modules corresponded to the hieratical differentiation neutrophil maturation in hematopoiesis, with M1 representing the start, and M5 representing the end of pseudotime in scRNA-seq (Fig 5C and 5D).

To identify the modules affected by MAP4K4 during neutrophil differentiation, we analyzed the pseudotime trajectory of Ctrl and cKO data. The differential module eigengene (DME) analysis revealed significant differences between Ctrl and cKO in all modules (Fig 5E), and M3 and M4 were the most affected by MAP4K4-deficient with the lowest distance correlation score (dor) (Fig 5E) followed by M2. Notably, genes in M2, M3, and M4 were all enriched in pathways related to apoptosis (S5F Fig and S11 Table). Furthermore, GSEA analysis revealed that 'Positive regulation of apoptotic process' pathway was significantly enriched in Ctrl for ProNeu (p.adjust = 0.049), cKO for PreNeu (p.adjust = 0.047), and cKO for Immature-Neu (p.adjust = 0.043), indicating that the process of apoptosis was inhibited in ProNeu of cKO, while promoted in PreNeu and imNeu of cKO (Fig 5F, **Methods**). We measured the 'apoptosis score' based on the averaged gene expression level in the 'positive regulation of apoptosis process' pathway (Fig 5G and S12 Table). We found that the apoptosis score for ProNeu cells decreased the most in cKO mice, in contrast, the apoptosis score increased in

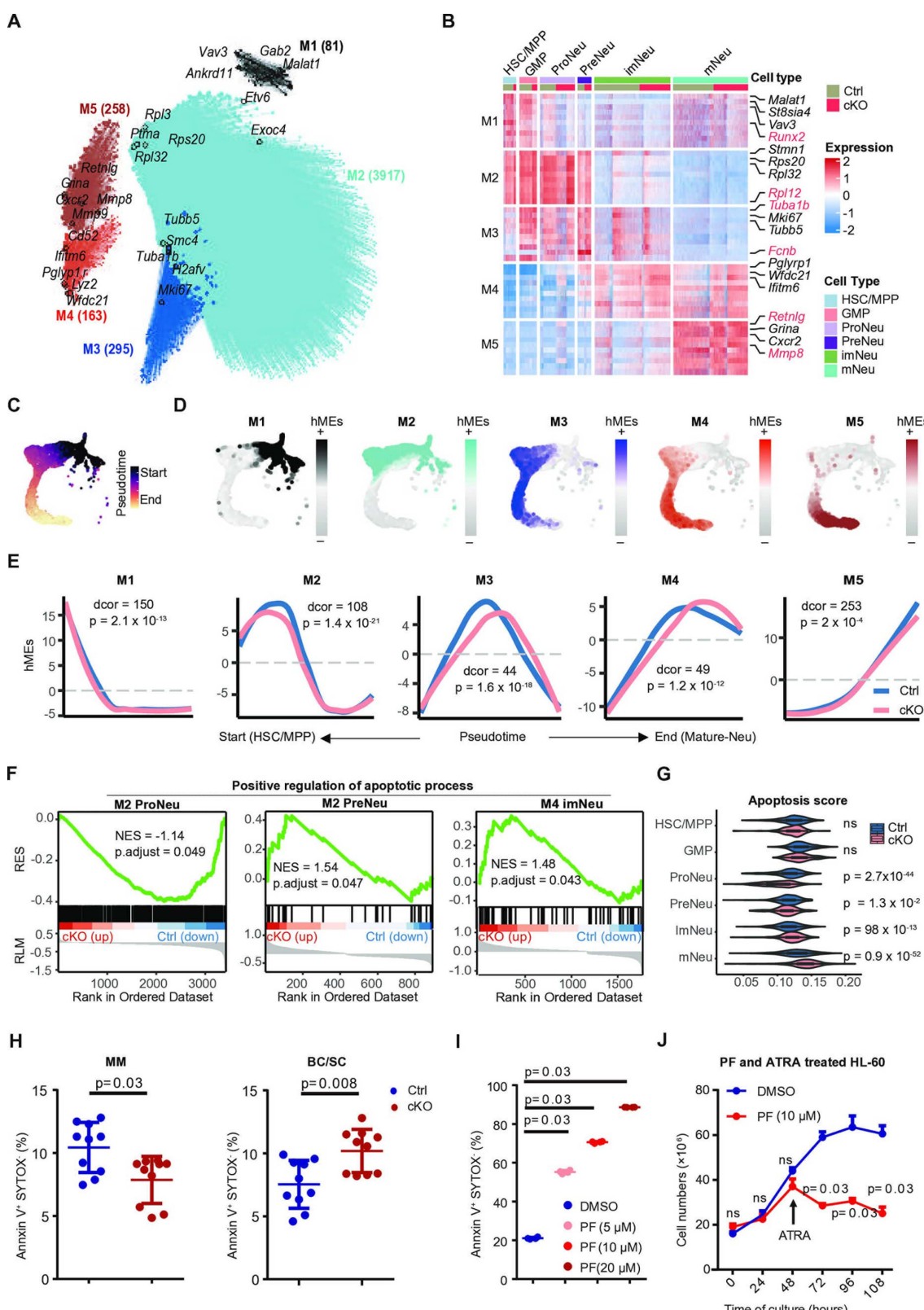

**Fig 5. Loss of MAP4K4 expression affects apoptosis during the process of neutrophil differentiation.** (A) UMAP plot of the neutrophil co-expression network. Each node represents a single gene, and edges represent co-expression links between genes

and module hub genes. Point size is scaled by eigengene-based connectivity (kME). Nodes are colored by co-expression module assignment. The top five hub genes per module are labeled. Network edges were down-sampled for visual clarity. (B) Heatmap of scaled gene expression for the top 10 hub genes by kME in each module. (C) UMAP colored by Pseudotime. (D) UMAP colored by harmonized module eigengenes (hMEs) of each module. (E) Module eigengenes (MEs) as a function of pseudotime for each co-expression module. For each module, a separate loess regression line is shown for each condition. The p value was calculated by distance correlation t-test (dcorT.test). dcor is a transformation of a bias-corrected version of distance correlation. The smaller the dcor, the greater the distance between control (Ctrl) and *Map4k4*-cKO (cKO) conditions. (F) Differentially expressed hub genes between control (Ctrl) and *Map4k4*-cKO (cKO) parasites involved in positive regulation of apoptotic process determined by Gene Set Enrichment Analysis (GSEA). RES represents the running enrichment score. RLM represents the ranked list metric. (G) Violin plots of apoptosis scores of control (Ctrl) against *Map4k4*-cKO (cKO) for each cluster, the p value was calculated by the Student's t-test. (H) Percentages of annexin V$^+$ and SYTOX$^-$ cells in bone marrow neutrophil progenitor cells of control (Ctrl) or *Map4k4*-cKO (cKO) mice; MM, metamyelocytes; BC/SC, band cells and segmented neutrophils; BM, bone marrow (n=10; mean ± SD). Mann-Whitney U test. (I) Percentages of annexin V$^+$ and SYTOX$^-$ cells of HL-60 cells treated with 0-20 μM MAP4K4 inhibitor (PF-06260933, PF) for 48 hours, followed by a subsequent 96-hour treatment with both PF and all-trans-retinoic acid (ATRA) (n=4; mean ± SD). Mann-Whitney U test. (J) Quantification of HL-60 cell numbers after a 48-hour treatment with a 10 μM concentration of the MAP4K4 inhibitor (PF-06260933, PF), followed by a subsequent 96-hour treatment with both PF and all-trans-retinoic acid (ATRA) (n=4; mean ± SD). Mann-Whitney U test.

cKO-imNeu and mNeu (Fig 5G). Apoptosis-related genes such as *Rpl11* and *Prelid1* upregulated in all cKO cell types from HSC/MPP to PreNeu except for ProNeu, while *Myc*, *Acin1,* and *Pdcd4* were only downregulated in cKO-ProNeu (S5G Fig). We validated these results by assessing neutrophil cell death rates at different differentiation stages using a combination of annexin V and SYTOX staining. This analysis indicated reduced apoptosis rates among cKO-promyelocytes (proNeu) [7], cKO-myelocytes (mixture of GMP and proNeu) [7], and cKO-metamyelocytes (preNeu) [7]. In contrast, apoptosis rates increased by 25% in cKO-band cells and segmented neutrophils (imNeu/mNeu) [7] (Figs 5H and S5H). These findings underscore the differential apoptotic responses across various stages of neutrophil maturation in the cKO model.

To further assess the role of MAP4K4 in regulating neutrophil apoptosis, we conducted *in vitro* studies using HL-60 cells. Notably, the treatment of MAP4K4 inhibitors (PF-06260933) in HL-60 cells did not increase cell apoptosis unless at a high (20 μM) concentration (S5I and S5J Fig). However, when the cells were treated with neutrophil-like state inducer (ATRA), MAP4K4 inhibitors resulted in cell apoptosis even at low (5 μM) concentration. And this apoptotic-inducing effect was dose-dependent. To confirm this phenotype, we treated the cells continuously with the inhibitor at a moderate (10 μM) concentration, 24 hours after treatment of ATRA, the cell numbers reduced by 50% compared to the control group (Fig 5I and 5J). These results demonstrate the significant impact of MAP4K4 on cell viability during the neutrophil differentiation process.

## MAP4K4 modulates the phosphorylation level of apoptosis-related genes

As a serine/threonine protein kinase, MAP4K4 plays a pivotal role in modulating protein phosphorylation [60], our *in silico* knockout also predicted it affects phosphorylation (Fig 2L). To investigate how MAP4K4 influences downstream protein phosphorylation levels, we performed mass spectrometric analysis on protein expression and phosphorylation level in a cell line lacking expression of MAP4K4 (Fig 6A). Most peptides ranged between 7 and 20 amino acids (S6A and S6B Fig) and the principal components analysis (PCA) results showed samples were clustered by sample types, indicating good sample quality (S6C and S6D Fig). Compared to the controls, the phosphorylation level of 1911 proteins, and the expression of 1606 proteins were significantly changed in *MAP4K4*-KO cells (Fig 6B and S13 and S14 Tables). Notably, *MAP4K4*-KO has more impact on protein phosphorylation level than the expression level (Fig 6B and 6C).

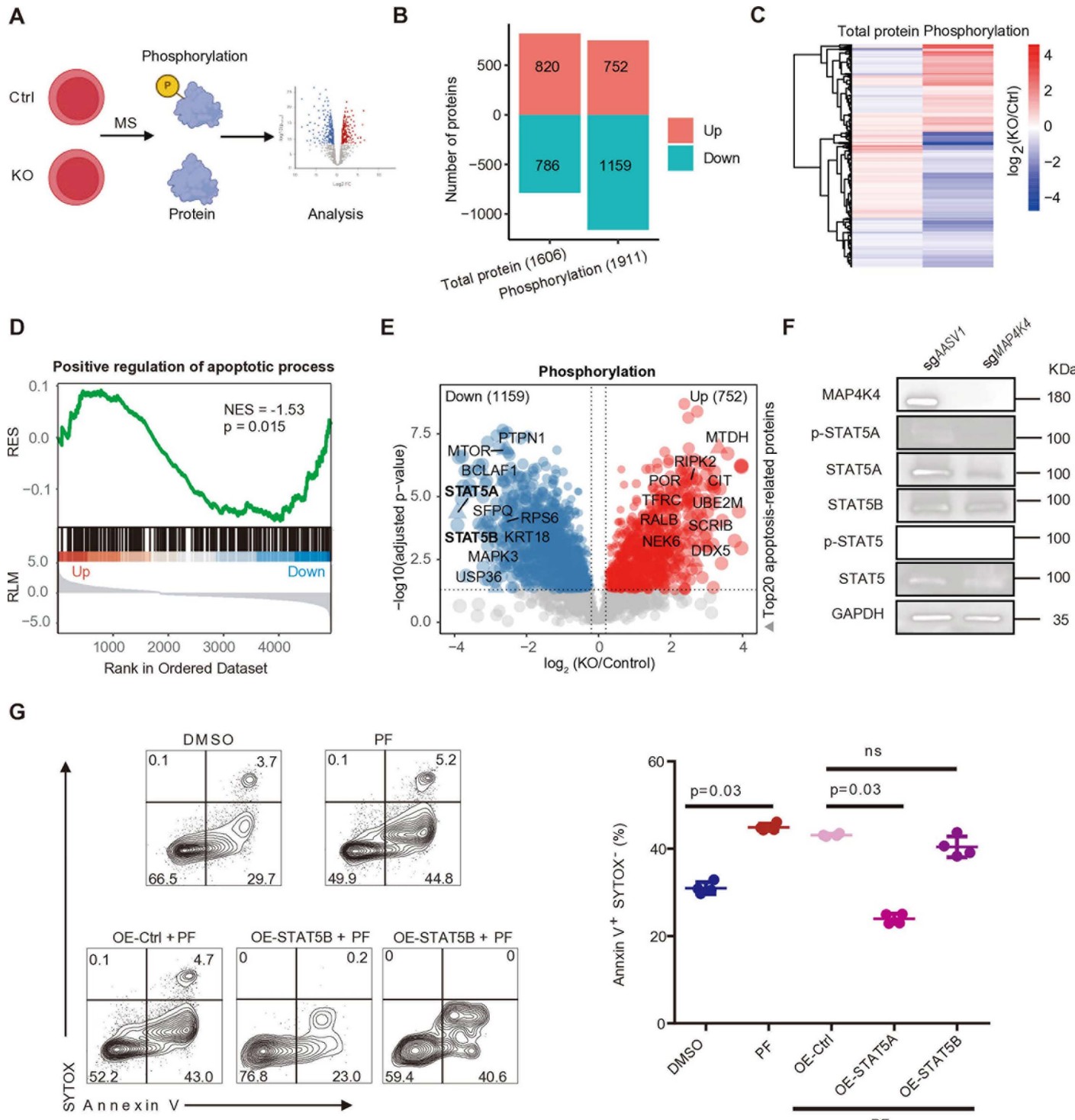

**Fig 6. Knocking out *MAP4K4* reduces phosphorylation levels in the apoptosis pathway, and overexpression STAT5A rescued MAP4K4 deficiency-caused neutrophil apoptosis.** (A) Schematic of mass spectrometry analysis, created using Biorender. (B) The number of proteins with significant differences in protein and phosphorylation levels after *MAP4K4* knockout. (C) Comparison heatmap of log $_2$ (fold-change) in protein and phosphorylation levels after *MAP4K4* knockout. (D) Differential phosphorylation level between control and *MAP4K4* KO parasites is involved in the positive regulation of the apoptotic process, as determined by Gene Set Enrichment Analysis (GSEA). RES represents the running enrichment score. RLM represents the ranked list metric. (E) Volcano plot of the apoptosis-related proteins with differentially expressed phosphorylation levels. The x-axis shows the phosphorylation difference of log$_2$ (fold-change) between *MAP4K4* KO and Ctrl. The y-axis shows –log$_{10}$ (adjusted p value) of the t-test between these two groups. Red indicates proteins with up-regulated phosphorylation after *MAP4K4*

KO, and blue, proteins with down-regulated phosphorylation after *MAP4K4* KO. The top 20 proteins with the most significant differences in phosphorylation among the apoptosis-related proteins are visualized. (F) Western blot (WB) for MAP4K4, p-STAT5A, STAT5A, p-STAT5B, STAT5B, and GAPDH in sg*AASV1* and sg*MAP4K4* cells. (G) Representative FACS analysis of HL-60 cells were first infected with overexpression of STAT5A or STAT5B or control plasmids and then treated with 10 μM MAP4K4 inhibitor (PF-06260933, PF) for 48 hours, followed by a subsequent 96-hour treatment with both PF and all-trans-retinoic acid (ATRA) for staining of anti–annexin V and SYTOX was shown (left). Percentages of annexin V⁺ and SYTOX⁻ cells (right) (n=4; mean ± SD). Mann-Whitney U test. Fig 6A was created with Biorender.

Then, we performed GO-BP analysis of differential phosphorylation level proteins between Ctrl and *MAP4K4*-KO samples. Notably, proteins with downregulated phosphorylation levels were specifically enriched with 'Positive regulation of apoptotic process' and 'Positive regulation of myeloid leukocyte cytokine production involved in immune response', while those up-regulated proteins were uniquely enriched with 'Mitochondrial ATP synthesis coupled electron transport' and 'Nucleotide phosphorylation' (S6E Fig). GSEA analysis showed that 'Positive regulation of apoptotic process' was significantly enriched for proteins with downregulated phosphorylation level (p.adjust = 0.015, Normalized Enrichment Score (NES) = -1.53) (Fig 6D). These results suggest that MAP4K4 may play a critical role in modulating phosphorylation levels of genes involved in apoptotic pathways, thereby influencing cell apoptosis during neutrophil differentiation.

Among the apoptosis-related proteins (S12 Table) [61] with downregulated phosphorylation levels, STAT5A and STAT5B exhibited the most downregulation level in phosphorylation with fold changes of 15 and 9, respectively (Fig 6E). We confirmed that both the protein expression and phosphorylation levels of STAT5A were significantly decreased in *MAP4K4*-KO k562 cells using Western blot analysis (Fig 6F), while the protein levels of STAT5B remained largely unaffected. Due to the absence of a specific antibody for phosphorylated STAT5B, its phosphorylation status could not be evaluated. To further delineate the distinct roles of STAT5A and STAT5B in the MAP4K4-mediated apoptosis during neutrophil differentiation, we overexpressed STAT5A, STAT5B, or control 3xFLAG proteins in HL-60 cells treated with MAP4K4 inhibitor (S6F Fig) and analyzed cell apoptosis level during ATRA-induced neutrophil differentiation. We found that overexpression of STAT5A but not STAT5B rescued the cell apoptosis induced by MAP4K4 inhibitor (Fig 6G). These findings suggest by regulating the phosphorylation level of apoptosis-related genes, such as STAT5A, MAP4K4 regulates apoptosis during the process of neutrophil differentiation.

## Discussion

By integrating recent developments in machine learning with the rapidly increasing amount of sequencing data, we trained a neutrophil-specific random forest model to predict all genes. We further validated our prediction using literature and experiment assays. The main strength of our approach is the following.

First, our model is designed for neutrophil differentiation, allowing for a more accurate prediction of functional genes involved in this process. It has long been appreciated that looking at the patterns of expression dynamics [62], physiological characteristics [63], pathological relatedness [64] and conservation [65] can yield insights into the consequences of functional genes within a specific biological process. So we integrated data from 21 features from the four categories, in which the expression profile was neutrophil-specific from FACS sorted bulk RNA-seq and pseudotime of scRNA-seq data.

Second, each machine-learning method has its pros and cons. We used various machine-learning models to solve problems at different stages. (1) PU learning [12], naïve Bayesian classification, and under-sampling for labeling. Conventional supervised learning

methods require both positive and negative labels. However, acquiring negative datasets is problematic because non-functional genes are largely undocumented. PU learning approach solved the problem by inferring potential negative genes from unlabeled genes and thus improving classifier performance. (2) Random forest for scoring [11], classification, and importance assessment. Random forest is widely used since it provides both high prediction accuracy and model interpretability. Unlike other models such as CellOracle [9] and SCENIC [10], limited to TFs, NeuRGI provides scores and classifications for all gene types, including enzymes, membrane proteins, and RBPs. (3) Deep-learning neural network using VAE for *in silico* knockout [13]. We used OntoVAE to predict the effect of the knockout of a gene, prioritizing regulatory genes and directing the follow-up functional experiments.

Third, our model demonstrates outstanding performance, achieving an AUC of 0.977. Specifically, it not only validates well-known TFs with high scores, such as *PU.1*, *GFI-1* and *CEBPA* (with NeuRGI scores of 0.9991, 0.9983, and 0.9976, respectively), but also predicts the novel gene *MAP4K4*, which was further experimentally validated. MAP4K4 plays a crucial role in T cell functionality and presents as a potential therapeutic target for enhancing LFA-1 activation in CD8 T cells, thereby augmenting their adhesion, priming, and cytotoxic activities to boost antitumor and antiviral immunity [16]. Concurrently, HGK deficiency in T cells induces the degradation of TRAF2, elevating IL-6 levels, which promotes Th17 differentiation and leads to insulin resistance [18]. We observed that HSPCs lacking MAP4K4 expression showed a reduced differentiation capacity into granulocytes and monocytes, while differentiation into erythrocytes and platelets remained unaffected. Through single-cell sequencing, we constructed a gene expression network during bone marrow neutrophil differentiation in MAP4K4-deficient mice. Our analysis reveals that MAP4K4 finely regulates the neutrophil differentiation process. Additionally, while STAT5A is typically phosphorylated by JAK tyrosine kinases [66] and is known for its role in anti-apoptotic signaling and enhancing cell survival [67–70], our novel findings suggest that MAP4K4 can also phosphorylate STAT5A, a function previously unreported in the literature. These findings illuminate the multifaceted role of MAP4K4 in immune cell differentiation and function, underscoring its potential as a therapeutic target in modulating immune responses. The ability of MAP4K4's newly discovered role in STAT5A phosphorylation opens new avenues for research into its mechanisms and therapeutic potential in immune-related disorders.

Nevertheless, we note some limitations of the method. Firstly, the classification of functional genes obtained using GMM is still arbitrary, given the limited number of positive and negative genes in the training set. Relying solely on NeuRGI scores for ranking may overlook some potentially important genes. Second, we identified many potential genes such as FIG4 and *TIGAR*, which are not been experimentally validated given the time and funding constraints. High-throughput CRISPR screening [71,72] may provide an intermedial solution to bridge this gap by further narrowing down function genes. Third, the specificity of the data used in this study makes NeuRGI tailored for the neutrophil in myeloid linage but not for other hematopoietic lineages, such as lymphoid and erythroid lineages. However, this limitation can be addressed by incorporating lineage-specific positive gene sets, along with features, such as single-cell epigenomics and proteomics [73,74], that capture the unique characteristics of each lineage. Thus, the paradigm of our study can be extended to other lineages or other tissues.

## STAR methods

### Ethics statement

All animal experiments followed the protocols approved by the Institutional Animal Care and Use Committee of West China Second University Hospital [(2018) Animal Ethics Approval No. 004].

## Overview of neutrophil regulatory gene identifier (NeuRGI)

To build a machine learning-based Neutrophil Regulatory Gene Identifier (NeuRGI) model, we used a workflow consisting of several steps: (1) Data collection. (2) Feature extraction. (3) Defining the positive and negative sets using PU-learning [75]. (4) 10-fold cross-validation [76]. (5) Model training. (6) Scoring and ranking. (7) Gaussian Mixture Model (GMM) [77] classified NeuRGI score. We implemented and tested our workflow in R (version 4.0), our code is open source and available on GitHub (https://github.com/LuChenLab/NeuRGI).

The framework of NeuRGI consists of 7 key steps:

**Step 1: Data collection and preprocessing.** For expression dynamics, we collected bulk RNA sequencing (bulk RNA-seq) data, including 8 cell types involved in neutrophil differentiation from hematopoietic stem cells (HSCs) isolated from bone marrow samples (mouse, GSE142216) [19] and umbilical cord blood (human, EGAD00001000745) [20]. Single-cell RNA sequencing (scRNA-seq) data from bone marrow neutrophils (mouse, GSE243466) [21] was also collected. For physiological characteristics and pathological relatedness, datasets were downloaded from databases DisGeNET [24], MsigDB [25]. For the conservation score, we collected the PhastCons file for the conservation of multi-species alignments which was downloaded from the UCSC Genome Browser [78]. Data preprocessing and more details were provided in the S1 Table.

**Step 2: Feature extraction.** We extracted 28 features of genes covering 4 aspects, including expression dynamics, physiological characteristics, pathological relatedness, and conservation score (S2 Table).

1. Expression dynamics

We extracted 16 features from bulk and scRNA-seq data separately. For mouse and human bulk RNA-seq data, the Spearman correlation, Spearman p value, range, slope, fit p value, and cell specificity Tau value were extracted. For mouse scRNA-seq data, the Spearman correlation, Spearman p value, range, and slope were extracted. The bulk RNA-seq represented the transcriptome from Fluorescence-activated Cell Sorting (FACS-sorted) cells, while the cell types of scRNA-seq were assigned by the expression of gene markers. For bulk RNA-seq [19,20], cell types were sorted based on lineage commitment and assigned consecutive numbers starting from 1 to represent pseudotime. For example, HSC, MPP, GMP, ProNeu, and PreNeu were set as 1, 2, 3, 4, and 5, respectively. For scRNA-seq, the R package monocle (v2.22.0) [47] was used to infer the pseudotime of 2,803 immature and mature neutrophils from the bone marrow dataset [21] with default parameters. Then features were extracted the same way from bulk RNA-seq and scRNA-seq separately using their respective pseudotime. Homologous genes between mice and humans were aligned using R package biomaRt (v2.46.3) [22]. The Mouse and Human BioMart databases were extracted using the useMart function, followed by the alignment of homologous genes using the getLDS function.

*Feature. Spearman.cor and Spearman.p*
The correlations between gene expression and pseudotime were calculated. The correlation (*Spearman.cor*) was calculated by the function cor with parameter 'method = spearman' from the R package stats, and the correlation test (*Spearman.p*) was performed using the function cor.test with parameter 'method = spearman, alternative = two.side' from the R package stats (v4.0.2).

*Feature. Range*
The range of value (Range) was the expression difference value between maximum and minimum values, which means the measures of max variation. The range was calculated as the

TPM values for each gene during neutrophil differentiation. Furthermore, the range of value was scaled to 0~1 by redefining the maximum value and labeled as Range.

***Feature. Slope and fit p value***

Linear regression analyses between gene expression and pseudotime were conducted. The regression coefficient (Slope), symbolizing the extent of influence of the independent variable on the dependent variable, and the p value of the regression variable (fit p value) indicating the statistical significance of this influence, were extracted from the regression model.

***Feature. Tau***

Cell specificity index (Tau) [79], is a quantitative, graded scalar measure of the specificity of an expression profile. Tau interpolates the range between 0 for housekeeping to 1 for strictly stage or cell-type-specific.

$$Tau = \frac{\sum_{i=1}^{N}\left(1-x_i\right)}{N-1},$$

where N is the number of neutrophil lineage commitment cell types, and $x_i$ is the gene expression normalized to the maximal expression.

2. Physiological characteristics

From Open Targets Platform [23], we collected single nucleotide polymorphisms (SNPs) associated with neutrophil-related physiological phenotypes includingneutrophil count, neutrophil measurement, neutrophil percentage of granulocytes, neutrophil percentage of leukocytes, neutrophil to lymphocyte ratio, and sum of neutrophil and eosinophil counts. Furthermore, we conducted a statistical analysis of the host genes for these SNPs. Genes with one SNP were labeled as 1, those with two SNPs were labeled as 2, and genes without SNPs were labeled as 0.

3. Pathological relatedness

First, we searched for neutrophil-related pathological phenotype 'abnormal neutrophil count' from MsigDB [25]. The values were transformed into binary form, where genes related to the pathological phenotype were labeled as significant (1), while the rest as nonsignificant (0).

Second, we collected disease features below from DisGeNET [24]

***Feature. DSI.***

The Disease Specificity Index (DSI) ranges from 0 to 1 and is inversely proportional to the number of diseases associated with a particular gene. A gene associated with a large number of diseases (e.g., TNF, associated with more than 1500 diseases) will have a DSI close to zero, while a gene associated with only one disease, is more 'specific' for that disease and has a DSI of 1.

***Feature. DPI***

The Disease Pleiotropy Index (DPI) ranges from 0 to 1 and is proportional to the number of different (MeSH) disease classes a gene is associated with. Thus, a gene associated with diseases of diverse classes (such as APOE, associated with Cardiovascular Diseases, Mental Disorders, Neoplasms, Respiratory Tract Diseases, etc), will have a DPI close to 1. Conversely, the PSCA, associated with 58 diseases, most of which are neoplasms has a relatively low DPI.

***Feature. No of diseases***

The number of diseases associated with the gene

***Feature. PLI***

The probability of the gene being loss-of-function intolerant (PLI) is provided by the gnomAD consortium. PLI closer to 1 indicates that the gene or transcript cannot tolerate protein truncating variation (nonsense, splice acceptor, and splice donor variation). The gnomAD

team recommends transcripts with a PLI>= 0.9 for the set of transcripts extremely intolerant to truncating variants. PLI is based on the idea that transcripts can be classified into three categories: 1) null: heterozygous or homozygous protein-truncating variation is completely tolerated; 2) recessive: heterozygous variants are tolerated but homozygous variants are not; 3) haploinsufficient: heterozygous variants are not tolerated. An expectation-maximization algorithm was then used to assign a probability of belonging in each class to each gene or transcript. PLI is the probability of belonging to the haploinsufficient class.

4. Conservation score

Mean phastcons was the only feature in the aspect, which represented the conservation of a gene across different species. The mean phastcons was calculated based on the conservation data phastCons100way.UCSC.hg38 of 99 vertebrates, with the data obtained from the UCSC Genome Browser (https://hgdownload.cse.ucsc.edu/goldenpath/hg38/phastCons100way/) [79].

**Step 3: Defining the positive and negative sets.** The positive set (P set) included 293 protein-coding genes with the Gene Ontology (GO) of keyword 'neutrophil' and reported in the literature [8,27,29], whereas the negative training set was confirmed by the PU-learning Spy algorithm [12]. In this approach, we randomly selected 10% of the genes (set parameter prob = c(0.9,0.1) in sample function) from the P set to act as "spies" and added them to the Unlabeled set (U set, including 19,581 genes). This created the Us set, which now contained 19,603 genes. The remaining genes from the P set formed the Ps set, which contained 271 genes. To balance the sizes of the sets, we performed oversampling on the Ps set (from 271 to 19,603 genes) using the ovun.sample function from the R package ROSE (v0.0-4) [80], matching the number of genes in the Us set. We then trained a Naive Bayes classifier (G) on the combined Ps+Us training set, Ps as positive and Us as negative, using the classifier G to predict probabilities for both the original U set and the spies. The 10th percentile of the probabilities for spies was used as a threshold, below which genes in the U set were classified as reliable negatives (RN).

**Step 4: 10-fold cross-validation.** The PU-learning algorithm obtained 8,393 genes in the negative set, but only 293 genes in the positive set. Therefore, we applied the under-sampling [81] method to balance the positive and negative sets of the training data. At the same time, we used stratified sampling to ensure that the number of each gene type in the negative set is consistent with that in the positive set. To reduce randomness, the dataset was processed with 10-fold cross-validation [76] to evaluate and improve the performance of the NeuRGI models, which involved dividing the data into ten distinct splits, each producing corresponding training and testing sets. Following the 10-fold cross-validation process, we constructed ten distinct datasets from the original data. Each of these datasets was specifically prepared to serve as independent training and testing sets, ensuring that every model training cycle was based on different segments of the data.

**Step 5: Model training.** Random forest classifiers were systematically trained on the collected datasets, utilizing the randomForest function of the R package randomForest (v4.6-14) [82]. The mtry parameter was configured to 6. The ntree parameter was configured to 1000. Each of the ten data splits, derived from a 10-fold cross-validation process, was employed to train the model. Subsequently, each model was rigorously validated against its corresponding validation set. This structured approach enhanced the reliability and predictive accuracy of the NeuRGI by ensuring each model was comprehensively tested across different subsets of data.

**Step 6: Scoring and ranking.** NeuRGI score for each gene was calculated by averaging the scores obtained from the ten random forest models, each developed through rigorous

training on distinct data splits. This calculation involved scoring each gene individually by each of the ten models, reflecting a comprehensive assessment of its potential metabolic relevance based on different subsets of the data. We utilized the NeuRGI model to predict all other genes not involved in the model training, which were ranked according to NeuRGI.

**Step 7: Gaussian mixture model (GMM) classified NeuRGI score.** The ranges of the two peaks in the NeuRGI score density were determined using the findPeaks function from the R package quantmod (v0.4.26) [32]. A Gaussian Mixture Model (GMM) was applied directly to the NeuRGI scores to characterize the distribution of predicted gene scores. The analysis was performed using the mclust package in R (v6.0.0) [77]. The GMM was fitted using the function Mclust with the parameters G = 3 and modelNames = 'V', which models the data as originating from three Gaussian distributions with variable variances for each cluster. The model is defined as follows:

$$p(x) = \sum_{3}^{k=1} \pi_k N(x\ \mu_k, \Sigma_k)$$

Where $\pi_k$ represents the mixture proportions, and $N(x\mu_k, \Sigma_k)$ is the Gaussian distribution with mean $\mu_k$ and covariance $\Sigma_k$ for each component $k$. The parameters $\pi_k$, $\mu_k$, and $\Sigma_k$ were estimated from the data using the Expectation-Maximization (EM) algorithm.

## Evaluation of NeuRGI performance

**1. Area under the receiving operator characteristic curve (AUC).** The area under the receiving operator characteristic curve (AUROC) was used to assess the performance of NeuRGI using the R package pROC (v1.18.5) [83]. The sensitivity and specificity of the model were extracted from the results of the roc function (**Fig 1B**).

$$\text{Sensitivity} = \frac{\text{TP}}{\text{TP} + \text{FN}}$$

$$\text{Specificity} = \frac{\text{TN}}{\text{TN} + \text{FP}}$$

$$AUC = \int_1^0 \text{Sensitivity}(1 - \text{Specificity})\, d(1 - \text{Specificity})$$

Where TP, FP, TN, and FN represent true positives, false positives, true negatives, and false negatives, respectively.

**2. Area under the precision-recall curve (AUC-PR).** The Area Under the Precision-Recall Curve (AUC-PR) is an alternative to AUC, particularly useful when dealing with imbalanced datasets. This metric evaluates the trade-off between precision and recall at different thresholds, focusing more on the positive class. The AUC-PR is the area under the Precision-Recall curve, which plots precision against recall. A higher AUC-PR value indicates better model performance in terms of precision and recall balance. We used pr.curve function from the R package PRROC (v1.3.1) to calculate AUC-PR (**Fig 1B**).

$$\text{Precision} = \frac{TP}{TP + FP}$$

$$\text{Recall} = \frac{TP}{TP + FN}$$

$$AUC - PR = \int_{1}^{0} \text{Precision}(\text{Recall}) d(\text{Recall})$$

**3. Accuracy (ACC).** Accuracy is the proportion of correct predictions out of the total number of predictions. It is calculated as the ratio of the sum of True Positives (TP) and True Negatives (TN) to the total number of instances. Accuracy is suitable when the dataset is balanced but may be misleading when the dataset is imbalanced.

$$\text{Accuracy} = \frac{TP + TN}{TP + TN + FP + FN}$$

**4. Matthews correlation coefficient (MCC).** The Matthews Correlation Coefficient (MCC) is a more balanced metric for evaluating binary classification performance, taking into account true positives, true negatives, false positives, and false negatives. The MCC value ranges from -1 (perfect disagreement) to +1 (perfect agreement), with 0 indicating random classification performance.

$$MCC = \frac{TP \cdot TN - FP \cdot FN}{\sqrt{(TP + FP)(TP + FN)(TN + FP)(TN + FN)}}$$

**5. F1 score.** The F1 Score is the harmonic mean of precision and recall, providing a single metric that balances both. It is particularly useful when dealing with imbalanced datasets, as it emphasizes both false positives and false negatives. The F1 Score is calculated as follows:

$$\text{F1Score} = 2 \times \frac{\text{Precision} \times \text{Recall}}{\text{Precision} + \text{Recall}}$$

## Feature importance

We applied the mean decrease coefficient in the Gini Coefficient [84] to evaluate the importance of the features. This coefficient indicates how well features split the data at each node in the trees, reflecting its substantial role in improving node purity and classification accuracy. Gini impurity reduction was calculated for each split in decision trees. The decrease in Gini from each feature was summed up across all nodes and trees. This sum was divided by the total number of splits of features, producing the Mean Decrease Gini index for each feature. The Mean Decrease Gini index was calculated as follows:

$$G(t) = 1 - \sum_{i=1}^{J} p_i^2$$

$$\Delta G = G(t) - \left[ \frac{n_L}{n} G(t_L) + \frac{n_R}{n} G(t_R) \right]$$

where G(t) represents the Gini impurity at node t, pi is the proportion of class i samples at node t, and J is the total number of classes. N is the number of samples at node t, and nL and nR are the number of samples at the left and right child nodes, respectively.

## Methods comparison with CellOracle and SCENIC

To compare NeuRGI with CellOracle [9] and SCENIC [10], 2,803 immature and mature neutrophils from mouse bone marrow scRNA-seq data were used [21].

**CellOracle.** Python package CellOracle (v0.20.0) [9] was used to predict functional TFs by simulating knockouts and calculating perturbation scores. First, the gene regulatory network (GRN) models were constructed using two types of input data including scRNA-seq data and a base GRN. Preprocessing of the scRNA-seq data was performed using the Python package Scanpy (v1.10.4) [85]. Specifically, the data were normalized using the function sc.pp.normalize_per_cell, and non-variable genes were removed by applying sc.pp.filter_genes_dispersion with parameters n_top_genes=2000 and log=False. This step aimed to improve the overall accuracy of GRN inference by filtering out noisy genes. Logarithmic transformation and scaling were applied using the functions sc.pp.log1p and sc.pp.scale, respectively. Dimensionality reduction was performed using the function sc.tl.pca. Cell annotations were assigned based on available metadata [21]. The base GRN, pre-built into CellOracle, was derived from the mouse sciATAC-seq Atlas dataset, which encompasses a wide range of tissues and cell types. GRN inference was conducted by integrating the preprocessed scRNA-seq data with the base GRN, utilizing the get_links function in CellOracle. Based on the GRN model, the effects of TF perturbations on cell identity were simulated using the simulate_shift function. Transition probabilities were subsequently computed with the estimate_transition_prob function. To compare the TF perturbation vector field generated by CellOracle with the developmental vector field, the inner product score was calculated using the calculate_inner_product function. The developmental vector field was reconstructed from pseudotime data, which was computed with the Monocle2 R package (v2.22.0) [47], as detailed in Step 2: Feature extraction.

The calculated Inner product was defined as the perturbation score (PS). A negative PS indicates that the corresponding TF perturbation inhibits differentiation, while a positive PS suggests that the perturbation promotes differentiation. To quantify the magnitude of a TF's impact on differentiation regardless of its direction, the absolute value of the PS was used. TFs were ranked in descending order of their absolute PS values, with higher scores signifying greater relevance in neutrophil differentiation. All parameters are set as default if not mentioned.

**SCENIC.** Python package pySCENIC (v0.12.1) [10] was used to identify functional TFs based on regulon activity. The preprocessed scRNA-seq data are directly sourced from the CellOracle [9] preprocessed scRNA-seq pipeline. The transcription factors list was downloaded from https://github.com/aertslab/pySCENIC, ranking databases were downloaded from https://resources.aertslab.org/cistarget/databases/mus_musculus/mm10/refseq_r80/mc_v10_clust/gene_based/ mm10_10kbp_up_10kbp_down_full_tx_v10_clust. genes_vs_motifs.rankings.feather, and motif annotations were downloaded from https://resources.aertslab.org/cistarget/motif2tf/motifs-v9-nr.mgi-m0.001-o0.0.tbl. First, coexpression modules between TFs and candidate target genes are inferred with function pyscenic grn. Second, RcisTarget identifies modules for which the regulator's binding motif is significantly enriched across the target genes and creates regulons with only direct targets using function pyscenic ctx. AUCell scores the activity of each regulon in each cell using function pyscenic aucell with parameter auc_threshold = 0.01. Finally, the mean regulon activity for all cells is calculated for each regulon and ranked in descending order. The greater the regulon activity, the more important it is in neutrophil differentiation.

## GO network analysis of predictive functional genes

GO-BP enrichment analysis of predictive functional genes was performed using enrichGO from R package clusterProfiler (v4.2.2) [86]. The top 100 significant (p.adjust < 0.05) GO terms were reserved to calculate the GO semantic similarity matrix using the function GO_similarity from

the R package simplifyEnrichment (v1.0.0) [87]. The similarity matrix was then clustered using the function simplifyGO with parameter method = 'kmeans' from R package simplifyEnrichment (v1.0.0). The GO network was finally visualized using Cytoscape [88] (v3.8.2).

## MAGMA

To screen for genes related to neutrophil counts, MAGMA [46] was used to compute gene p value using GWAS summary statistics data. Firstly, we used a reference panel based on individuals of European ancestry in the 1000 Genomes Project [89]. Then we used a 10-kb window around the gene body to map SNPs to genes. And GWAS summary statistics 'neutrophil count' (ebi-a-GCST004629) was downloaded from the European Bioinformatics Institute (EBI) database (https://www.ebi.ac.uk/gwas/downloads/summary-statistics). GWAS summary statistics were next used as input for MAGMA (v 1.10) [46] to compute the Z score with default parameters.

## Predictive modeling of genetic perturbations using biologically informed variational autoencoders

See S1 Text.

## Mice

The C57BL/6 wildtype mice and $Map4k4^{f/-}$ mice were purchased from Gem Pharmatech Co. Ltd. $Map4k4^{f/-}$ mice were backcrossed against C57BL/6 mice, then intercrossed to obtain $Map4k4^{f/f}$ mice. The Chen Chong laboratory kindly provided mx1-cre mice and similarly backcrossed onto C57BL/6. $Map4k4^{f/f}$ mice were mated to Mx1-Cre $Map4k4^{f/f}$ mice to generate Mx1-Cre $Map4k4^{f/-}$ mice, then intercrossed to obtain Mx1-Cre $Map4k4^{-/-}$ (control, Ctrl) and Mx1-Cre $Map4k4^{f/f}$ ($Map4k4$-cKO, cKO) littermates for phenotypic analysis. Floxed alleles were deleted in Mx1-Cre $Map4k4^{f/f}$ mice 5-6 weeks before experiments by five intraperitoneal injections (300μg per mouse) of poly I:C (Sigma, #P1530-100MG) in PBS every other day. Mice were housed and bred under SPF conditions (Specific Pathogen Free) at the West China Second University Hospital Laboratory Animal Center. They were allowed access to diet and water *ad libitum.* All animal experiments followed the protocols approved by the Institutional Animal Care and Use Committee of West China Second University Hospital [(2018) Animal Ethics Approval No. 004].

## Cell culture

K562, HL-60, and HEK293T cell lines were initially purchased from the American Type Culture Collection (ATCC). K562 cells were maintained in IMDM medium (Gibco, #C12440500BT) containing 10% fetal bovine serum (FBS, YEASEN, #40130ES76) and 1% penicillin-streptomycin (Hyclone, #SV30010). HEK293T cells were maintained in DMEM medium (Gibco, #C11995500BT) supplemented with 10% FBS and 1% penicillin-streptomycin. HL-60 cells were maintained in RPMI 1640 medium (Gibco, #C11875500BT) supplemented with 10% FBS and 1% penicillin-streptomycin. HL-60 was induced by supplementing 1 μM all-trans-retinoic acid (MCE, #HY-14649).

## Viral production and transduction

The human STAT5A and STAT5B cDNA sequences were synthesized by Sangon Biotech and were reconstituted into the mammalian expression plasmid Plenti-EF1a-MCS-FLAG-6×His-CMV-mCherry-Puro (MiaoLing Biology, P59992). For human genes, sgRNAs targeting *MAP4K4* or sg*AAVS1* as control were connected into Plenti-CRISPR V2, donated by Wang Yuan Laboratory, via the Bsmb1 restriction site.

## Viral transduction

HEK293T cells were seeded at $6 \times 10^6$ cells per 10 cm culture dish and maintained in high-glucose DMEM (Gibco, #C11995500BT) supplemented with 10% FBS for 12–24 hours (h). Subsequently, a co-transfection was performed with 12 μg of the lentiviral construct (sg*AASV1* and 6 μg of psPAX2, 3 μg of pMD2.G packaging plasmid, using Liposomal Transfection Reagent (YEASEN, #40802ES03) according to the manufacturer's instructions. After 48 h of transfection, the supernatants were collected, and the culture was replaced with a fresh medium. Supernatants were again collected at 72 h after transfection. The collected supernatants were pooled, filtered through a 0.45 μm strainer, and centrifuged at 1000 g at 4 °C for 90 min. The concentrated viral particles were added to the cultured K562 cells and maintained for 36–48 h.

## Western blotting

K562 cells were lysed in RIPA lysis Buffer (Thermo Fisher Scientific, #89900) containing 10mM Protease Inhibitor Mini Tablets, EDTA-free (Thermo Fisher Scientific, #A32955) and phosphatase inhibitors (Beyotime, P1045) on ice for 30 min and centrifuged. Protein content in the supernatant was quantified using a Pierce BCA Protein Assay Kit (Thermo Fisher Scientific, #23227). Protein extracts were separated by 10% SDS polyacrylamide gel electrophoresis (PAGE) and transferred to nitrocellulose membranes (Millipore, #ISEQ00010). The blots were probed with the primary antibodies including rabbit anti-MAP4K4 antibody (CST, #3485), rabbit anti-p-STAT5A (Tyr699) antibody (Abcam, # ab32043), rabbit monoclonal (E289) anti-STAT5A antibody (Abcam, #ab32043), rabbit anti-STAT5B antibody (Abcam, #ab30648), rabbit monoclonal (C11C5) anti-p-STAT5 (Tyr694) antibody (CST, #9359), rabbit monoclonal (D2O6Y) anti-STAT5 antibody (CST, #94205), rabbit monoclonal (14C10) anti-GAPDH antibody (Cell Signaling Technology, #2118) in the universal antibody diluent (NCM biotech, #WB500D) for overnight at 4°C, washed three times with TBST, and then incubated with the HRP conjugated goat anti-rabbit antibodies (HuaBio, #HA1001). The chemiluminescence signal was detected using a ChemiDOCTMMP Imaging System (Bio-Rad Laboratories).

## Single-cell suspension

Bone marrow cells (BMCs) were collected from the tibia, femur, and ilium, lysed with red blood cell lysis buffer (Solarbio, #R1010), washed with PBS, and filtered through a 70 μm cell strainer (biosharp, #BS-70-CS).

Place the spleen on a 70 μm cell strainer (biosharp, #BS-70-CS), and use a sterile syringe plunger to gently break apart the spleen tissue. Rinse the strainer with PBS to collect the cells into a tube. Remove the supernatant, treat with a hemolytic solution to lyse red blood cells, then wash and resuspend the cells in PBS.

## Cell collection, RNA extraction, and qPCR

RNA was extracted using a Cell Total RNA Isolation Kit (FORE GENE, #RE-03113). Complementary DNA (cDNA) was synthesized using the RevertAid first-strand cDNA synthesis kit (Thermo Scientific, #K1621). PCR was performed using the 2 × Phanta Max Master Mix (Vazyme, P515). Real-time qPCR reactions were performed using the Blue qPCR SYBR Master Mix (YEASEN, #11184) on a Bio-Rad (CFX Connect) thermocycler. Primers were listed in the S15 Table.

## Flow cytometry assay

Single-cell suspension of BMCs, splenocytes, peripheral blood cells, or HL-60 cells were incubated with Fixable Viability Dyes (FVS, Biosciences) in PBS at room temperature in the dark for 20 minutes. The cells were next washed with staining buffer and incubated with

fluorochrome-conjugated antibodies listed below in staining buffer at 4 °C for 30 minutes. After thorough washing to remove excess antibodies, BMCs, splenocytes, and peripheral blood cells were analyzed using an Aurora CS Flow Cytometer (CYTEK), and Hl-60 cells were analyzed using an Attune Nxt (Life Technologies) flow cytometer. The data were analyzed using FlowJo software (BD).

To characterize hematopoietic stem and progenitor cells, cells were stained with BV785-conjugated anti-Sca-1 (BD Biosciences, #563991), BV711-conjugated anti-CD16/32 (BioLegend, #101337), BV605-conjugated anti-CD71 (BD Biosciences, #563013), BV421-conjugated anti-CD135 (BioLegend, #135315), PerCP-Cy5.5-conjugated anti-CD34 (Bio-Legend, #119328), PE-CF594-conjugated anti-CD127 (BioLegend, #135315), PE-conjugated anti-CD3e (BD Biosciences, #553062), PE-conjugated anti-Gr-1 (BioLegend, #108408), PE-conjugated anti-CD11b (BD Biosciences, #557397), PE-conjugated anti-B220 (BioLegend, #103208), PE-conjugated anti-TER119 (BD Biosciences, #553673), APC-Cy7-conjugated anti-CD117 (Biolegend, #105826), FITC-conjugated anti-CD41 (BD Biosciences, #553848).

To characterize bone marrow neutrophils, cells were stained with PerCP-Cy5.5-conjugated anti-CD34 (BioLegend, #119328), APC-Cy7-conjugated anti-CD117 (Biolegend, #105826), PE-Cy7-conjugated anti-Ly6G (BioLegend, #127618), PE-conjugated anti-CD3e (BD Biosciences, #553062), PE-conjugated anti-B220 (BioLegend, #103208), PE-conjugated anti-TER119 (BD Biosciences, #553673).

To characterize innate immune cells, BMCs were stained with BV711-conjugated anti-CD11b (Biolegend, #101242); BV605-conjugated anti-Ly6C (Biolegend, #128036), BV421-conjugated anti-F4/80 (Biolegend, #123137), APC-Cy7-conjugated anti-MHCII (BioLegend, #107628), PE-Cy7-conjugated anti-Ly6G (BioLegend, #127618), PE-conjugated anti-TER119 (BD Biosciences, #553673), PE-conjugated anti-CD3e (BD Biosciences, #553062), PE-conjugated anti-19 (BioLegend, #115508), PE-conjugated anti-Nkp46 (BioLegend, #137604), PE-conjugated anti-CD117 (BioLegend, #105808). For splenocytes, cells were stained with the same antibody panel except CD117. For peripheral blood cells, cells were stained with BV711-conjugated anti-CD11b (Biolegend, #101242), PE-Cy7-conjugated anti-Ly6G (BioLegend, #127618), PE-conjugated anti-TER119 (BD Biosciences, #553673). HL-60 cells were stained with PerCP-Cy5.5-conjugated anti-CD11b (BD Biosciences, #561114).

## Apoptosis assay

Single-cell suspension of BMCs or HL-60 cells was washed with PBS and subsequently resuspended in 1× Annexin V Binding Buffer (BD Biosciences, #556454) at a concentration of $1 \times 10^6$ cells/mL. The cell suspension (400 μL) was treated with Annexin V-BV421 (1:100 dilution; BD Biosciences, #563973) and SYTOX Green Dead Cell Stain (1:1000 dilution; Invitrogen, #S34860). Following incubation at room temperature for 30 min in the dark, the stained cells were immediately subjected to flow cytometric analysis.

## Neutrophil phagocytosis

E. *coli OP50-GFP* was cultured overnight at 37°C, then washed in PBS and counted. A suspension of *E. coli* was prepared to a concentration of $10^6$ colony-forming units (CFU) and was incubated with $5 \times 10^5$ neutrophils in a total of 1000 μL of RPMI media. The mixture was rotated at 8 rpm at 37°C for 30 minutes. Following the incubation, the sample was transferred to ice to halt phagocytosis. Neutrophils were then collected and stained with BV711-conjugated anti-CD11b (Biolegend, #101242) and PE-Cy7-conjugated anti-Ly6G (BioLegend, #127618) at 4°C. After washing with cold PBS 3 times, the phagocytosis percentages of gated CD11+ Ly6G+ cells were determined using a FACS scanner.

### Reactive oxygen species

A single-cell suspension of BMCs, spleen, or peripheral blood cells was incubated with 1 μM CellROX reagent (A) at 37°C in a 5% CO₂ atmosphere for 1 hour (Thermo Fisher Scientific, #C10492). During the final 15 minutes of incubation, 1 μL of 5 μM SYTOX Red Dead Cell stain solution (B) was added per 1 mL of the suspension. The stained cells were immediately analyzed by flow cytometry. Neutrophils were identified using anti-CD11b and anti-Ly6G monoclonal antibodies.

### Colony-forming unit assay

Bone marrow cells ($10^5$) were plated in methylcellulose culture medium (MethoCult M3434, STEMCELL Technologies) and incubated at 37°C with 5% $CO_2$. Colonies derived from CFU-E were counted at 48 h after plating, and the colonies derived from primitive erythroid progenitor cells (BFU-E) and granulocyte-macrophage progenitor cells (CFU-GM, CFU-G, and CFU-M) were counted 10-14 days after plating.

### Single-cell RNA-sequencing data analysis of HSPCs of Ctrl and *Map4k4*-cKO mice

**Library preparation and sequencing.** Freshly harvested bone marrow cells (BMCs) from *Map4k4*-cKO and control mice were isolated using HBSS without Mg2+/Ca2+, supplemented with 10 mM HEPES at pH 7.2. Subsequently, the BMCs from control or *Map4k4*-cKO mice were respectively divided into two aliquots. One aliquot underwent treatment with red blood cell lysis buffer to remove erythrocytes, while the other was subjected to enrichment for c-Kit+ BM hematopoietic stem and progenitor cells (HSPCs) using magnetic bead separation. The isolated c-Kit+ BM HSPCs were then artificially mixed with the corresponding BM cells at a ratio of 2:3 to establish a combined c-Kit/BM cell population. The library was performed according to the manufacturer's instructions (single cell 3' protocol, BMKMANU DG1000). GCs were processed to create cDNA with unique identifiers and barcodes, which were then purified, amplified, and prepared for sequencing with specific primers and indices during library construction. The libraries were sequenced on the Illumina NovaSeq 6000 platform at Biomarker Technologies (Beijing, China). On average, 110 Gb of raw data were generated for *Map4k4*-cKO samples and 89 Gb for control samples.

**Alignment, quantification, clustering, and annotation.** The raw reads were aligned to the mouse genome sequence (GRcm38) and gene quantified using CellRanger (v5.0.0). R package Seurat (v4.3.0) [58] was used for the clustering and annotation. Anchor genes between Ctrl and *Map4k4*-cKO samples were first identified by the function FindIntegrationAnchors (normalization.method = 'LogNormalize' and reduction = 'rpca') of R package Seurat (v4.3.0) [58]. These anchor genes represent genes whose expression profiles are shared between the samples, allowing us to align them and remove systematic differences caused by batch effects. With these anchors, we integrated the Ctrl and *Map4k4*-cKO samples using the IntegrateData function, which merges the datasets while preserving biological variation and correcting for unwanted batch effects. Cells of the integrated dataset were filtered using the following criteria: (a) feature counts below 200 or above 9000, (b) percentage of mitochondrial genes above 30, (c) identified as doublets using DoubletFinder [90] (v2.0.3) with the parameter of 'PCs = 1:15, pN=0.25, pK=0.09, nExp=24'. 9051 Ctrl cells and 5674 *Map4k4*-cKO cells were kept for further analyses. The integrated dataset was scaled using the function Scale Data. Then, we used principal component analysis (PCA) with variable genes as input and identified the top 16 significant PCs used as input for UMAP (Uniform Manifold Approximation and Projection). Cell markers from previously published studies [7,21] were

used to identify the cell types. Cell composition was calculated using the number of cells of each type divided by the total number of cells in this sample, the adjusted p value was calculated by Bonferroni corrected proportion test.

**Identification of differentially expressed genes.** We used the FindMarkers function (test.use = 't', logfc.threshold = log (1.5)) based on normalized data to identify differentially expressed genes (DEGs). P value adjustment was performed using Bonferroni correction based on the total number of genes in the dataset. DEGs with p.adjust > 0.05 were filtered out.

**Single-cell trajectory analysis.** Pseudotime trajectory analysis was conducted using the R package monocle (v2.22.0) [47] according to the tutorial with default parameters. Of the 14,725 cells in the WT and *Map4k4*-cKO integrated dataset, 11031 neutrophil-related cells including Hematopoietic stem cells/multipotent progenitor (HSC/MPP), granulocyte monocyte progenitor (GMP), neutrophil progenitors (ProNeu), neutrophil precursors (PreNeu), immature neutrophil (imNeu) and mature neutrophil (mNeu) were selected for the trajectory analysis.

## Co-expression analysis of neutrophils in Ctrl and *Map4k4*-cKO samples

Co-expression analysis was carried out using the R package hdWGCNA (v0.2.2) [59] and the R package WGCNA (v1.71) [91]. 11031 neutrophil-related cells including HSC/MPP, GMP, ProNeu, PreNeu, imNeu, and mNeu were selected for co-expression network analysis. We retained 54838 genes which were expressed in at least 5% of cells from any cluster. Metacell transcriptomic profiles were constructed separately for each of the 2 conditions and each cell type using the hdWGCNA function MetacellsByGroups with parameters 'k = 25, max_shared = 10, min_cells = 0'. We selected a soft-power threshold β = 9 based on the parameter sweep performed with the TestSoftPowers function. The co-expression network was computed with the ConstructNetwork function with the following parameters: networkType= 'signed', TOMType= 'signed', soft_power=9, deepSplit=4, detectCutHeight=0.995, minModule-Size=50, mergeCutHeight=0.2. Module eigengenes were computed using the ModuleEigen-genes function, and we applied Harmony to correct MEs based on the sequencing batch. Eigengene-based connectivity for each gene was computed using ModuleConnectivity. The co-expression network was embedded in two dimensions using UMAP with the RunMod-uleUMAP function with the top five genes (ranked by eigengene-based connectivity) per module as the input features. Differential condition module eigengene analysis was performed using R package energy (v1.7.7). The p value was calculated by the function dcorT.test. dcor is a transformation of a bias-corrected version of distance correlation calculated by the function dcorT. Hub genes of each module were extracted using the function GetHubGenes. Apoptosis score [92] was calculated using the function AddModuleScore from the Seurat R package [58] (v4.3.0) with apoptotic-related gene sets including GO:0043065 (S12 Table).

## Protein and protein phosphorylation mass spectrometry analysis

Approximately 1 x $10^8$ *MAP4K4* KO K562 cells, or control cells, were harvested to facilitate the identification and quantification of proteins and protein phosphorylation using proteomics mass spectrometry.

**Protein mass spectrometry analysis.** The tryptic peptides were dissolved in solvent A, and directly loaded onto a homemade reversed-phase analytical column (25 cm length, 100 µm i.d.). The mobile phase consisted of solvent A (0.1% formic acid, 2% acetonitrile/in water) and solvent B (0.1% formic acid, 90% acetonitrile/in water). Peptides were separated with the following gradient: 0-68 min, 6%-23%B; 68-82 min, 23%-32%B; 82-86 min, 32%-80%B; 86-90 min, 80%B, and all at a constant flow rate of 500 nl/min on an EASY-nLC 1200 UPLC

system (ThermoFisher Scientific). The separated peptides were analyzed in Orbitrap Exploris 480 with a nano-electrospray ion source. The electrospray voltage applied was 2300 V. FAIMS compensate voltage (CV) was set as -45 V, -65 V. Precursors, and fragments were analyzed at the Orbitrap detector. The full MS scan resolution was set to 60000 for a scan range of 400-1200 m/z. The MS/MS scan was fixed first mass as 110 m/z at a resolution of 15000 with the TurboTMT set as off. Up to 25 of the most abundant precursors were then selected for further MS/MS analyses with 20 s dynamic exclusion. The HCD fragmentation was performed at a normalized collision energy (NCE) of 27%. The automatic gain control (AGC) target was set at 100%, with an intensity threshold of 50000 ions/s and a maximum injection time of Auto.

**Protein phosphorylation mass spectrometry analysis.** The tryptic peptides were dissolved in solvent A, and directly loaded onto a homemade reversed-phase analytical column (25 cm length, 100 μm i.d.). The mobile phase consisted of solvent A (0.1% formic acid, 2% acetonitrile/in water) and solvent B (0.1% formic acid, 90% acetonitrile/in water). Peptides were separated with the following gradient: 0-70 min, 3%-20%B; 70-82 min, 20%-30%B; 82-86 min, 30%-80%B; 86-90 min, 80%B, and all at a constant flow rate of 500 nl/minon an EASY-nLC 1200 UPLC system (ThermoFisher Scientific). The separated peptides were analyzed in Orbitrap Exploris 480 with a nano-electrospray ion source. The electrospray voltage applied was 2300 V. FAIMS compensate voltage (CV) was set as -65 V, -45 V. Precursors, and fragments were analyzed at the Orbitrap detector. The full MS scan resolution was set to 60000 for a scan range of 400-1200 m/z. The MS/MS scan was fixed first mass as 110 m/z at a resolution of 30000 with the TurboTMT set as off. Up to 15 of the most abundant precursors were then selected for further MS/MS analyses with 30 s dynamic exclusion. The HCD fragmentation was performed at a normalized collision energy (NCE) of 27%. The automatic gain control (AGC) target was set at 75%, with an intensity threshold of 20000 ions/s and a maximum injection time of 100 ms.

**Selection of proteins with different protein expression or protein phosphorylation levels after MAP4K4 KO.** The fold change (FC) is defined as the ratio of the mean relative quantification values between *MAP4K4*-KO and Ctrl samples. The $\log_2$FC is calculated as follows:

$$\log_2\left(FC_{KO/Ctrl,k}\right) = \log_2\left(\frac{\text{Mean}\left(R_{ik}, i \in KO\right)}{\text{Mean}\left(R_{ik}, i \in Ctrl\right)}\right)$$

Where R represents the relative quantification value of the modified site, i denotes the sample, and k represents the modified site. A t-test was performed to test the significance. Proteins with $P < 0.05$ and abs($\log_2$FC) $> 0.2$ were identified as differentially expressed proteins. Phosphorylation sites with $P < 0.05$ and abs($\log_2$FC) $> 0.2$ were identified as differentially expressed phosphorylation sites. Since a protein may have multiple differentially phosphorylated sites, the change in the phosphorylation level of a protein is represented by the site with the largest change.

## GO-BP and GSEA analysis

GO-BP was performed using enrichGO and compareCluster from R package clusterProfiler [86] (v4.2.2). Gene Set Enrichment Analysis (GSEA) was performed using the function GSEA of R package clusterProfiler (v4.2.2) [86] with pvalueCutoff = 1. Genes were initially sorted based on $\log_2$ (FC). The enrichment was then performed using the GSEA function. The pvalueCutoff parameter was set to 0.5, and the cutoff of the p value was 0.05. Visualization was achieved with function gseaplot2 from the R package enrichplot (v1.10.2) [93]. Gene sets used to perform enrichment analysis were downloaded from org.Mm.e.g.,db (v3.12.0) package.

## Statistical analysis

Statistical analyses were performed by the R software 4.0.2 (https://www.R-project.org/). One-tailed Wilcoxon test was used to identify pathways that were significantly dysregulated upon knockout by OntoVAE. The two-sided Student's t-test was used to identify significantly differentially expressed genes in scRNA-seq data. Distance correlation t-test was used for hdWGCNA analysis. The two-sided Student's t-test was performed to identify proteins with different protein expression or protein phosphorylation levels after *MAP4K4* KO. For *in vivo* and *in vitro* experiments, data were presented as the mean ± SD of at least four independent experiments. The Mann-Whitney U test was utilized to analyze differences between experimental groups, except the RT-qPCR data, which were analyzed using a two-sided Student's t-test. The result with p value < 0.05 was considered statistically significant unless otherwise specified.

## Supporting information

**S1 Text. Description of *in silico* knockout process by OntoVAE.**
(DOCX)

**S1 Table. Dataset summary.**
(XLSX)

**S2 Table. Features used to train the model.**
(XLSX)

**S3 Table. Known positive and PU-learning negative training gene sets of NeuRGI.**
(XLSX)

**S4 Table. Feature ablation studies.**
(XLSX)

**S5 Table. 2,569 predicted functional genes with NeuRGI scores, significance of in silico knockout by OntoVAE and MAGMA Zscore.**
(XLSX)

**S6 Table. Celloracle and SCENIC predicted functional TFs during neutrophil differentiation.**
(XLSX)

**S7 Table. Top 12 genes with cell specificity index Tau, expression correlation coefficient, the significance of in silico knockout by OntoVAE, and NeuRGI score.**
(XLSX)

**S8 Table. Single-cell RNA sequencing quality control.**
(XLSX)

**S9 Table. Signature genes (top 30 DEGs) of each cell type.**
(XLSX)

**S10 Table. Top 10 differentially expressed genes after *Map4k4*-cKO.**
(XLSX)

**S11 Table. hdWGCNA module hub genes.**
(XLSX)

**S12 Table. List of apoptosis-related genes.**
(XLSX)

**S13 Table. Protein quantification by MS.**
(XLSX)

**S14 Table. Protein phosphorylation level quantification by MS.**
(XLSX)

**S15 Table. Resources and reagents.**
(XLSX)

**S1 Fig. NeuRGI model and OntoVAE model performance.** (A) Gene Ontology Biological Processes (GO-BP) enrichment of 293 PU-learning negative genes. These genes are not related to any neutrophil pathway. (B) Dot plot showing the feature importance of the baseline model using Gini coefficient with all features included. (C) Feature ablation studies. The line plots depict changes in the model's evaluation metrics—AUC, AUC-PR, ACC, MCC, F1 Score, and Mean (average of these five metrics)—as features are sequentially removed in reverse order of their importance. The results indicate that the model achieves optimal performance when the last seven features are excluded. (D) Density distribution of NeuRGI scores across different gene categories, including predicted function (red, 4,786 genes), predicted non-function (blue, 4,734 genes), predicted uncertain (grey, 9,768 genes), known positive (light red, 293 genes), and PU-learning negative (purple, 293 genes). (E) Pie chart illustrating the NeuRGI classification of 70 CellOracle-predicted TFs and 36 SCENIC-predicted TFs. (F) The boxplot illustrates the transformed feature values for 70 CellOracle-predicted TFs (up) and 36 SCENIC-predicted TFs (down) in four feature groups. The p value was calculated using the Student's t-test. (G) Heatmap illustrates the results of Kmeans clustering applied to the top 100 GO-BP terms associated with a set of 4,786 predictive functional genes. Rows and columns represent unique GO terms, with color intensity indicating similarity, where darker red denotes greater similarity. (H) The trend of test loss in OntoVAE model training. After 300 epochs of model training, the loss value no longer decreases and remains stable. (I) Scatter plot showing example pathway activity scores retrieved from OntoVAE model. The 'neutrophil activation' pathway is especially active in neutrophils and the 'regulation of mononuclear cell proliferation' pathway is especially active in monocytes.
(PDF)

**S2 Fig. Characteristic similarities of top genes.** (A) In silico knockout of ELANE and SYK in neutrophils and B cells, respectively. Bar plots display the affected neutrophil and B cell related terms ranked by significance. The dotted line represents the significance threshold. (B) The dot plot showed the NeuRGI score of the top 12 genes. Genes displayed in bold indicate those with limited reports on their involvement in neutrophil differentiation or function. (C) Expression of *MAP4K4*, FIG4, and *TIGAR* in different immune cells from ImmuNexUT. The number behind genes represents cell specificity Tau value. (D) Bar plot shows main gene pathways affected by OntoVAE *in silico* KO of *MAP4K4*, FIG4, and *TIGAR* in neutrophils. In the 'positive regulation of myeloid leukocyte differentiation' pathway, *MAP4K4* exhibits the lowest p value among the three genes. The dotted line represents the significance threshold. (E) Expression of *MAP4K4*, FIG4, and *TIGAR* in neutrophil differentiation. We set 'time cut' for cells at different differentiation stages, with HSC set as 1 and Neu as 5, and performed linear regression fitting for the expression of these 3 genes. R represents the Pearson correlation coefficient, and the p value was calculated by t-test.
(PDF)

**S3 Fig. Loss of MAP4K4 expression does not affect the number of hematopoietic stem and progenitor cells, erythrocytes, and platelets.** (A) Map4k4-deficient mice model. (B)

RT-qPCR analysis of *Map4k4* knockdown efficiency in mice BM cells (n=4; mean ± SD). Unpaired Student's t-test. (C) Bone marrow cell numbers control (Ctrl) or *Map4k4*-cKO (cKO) mice; BM, bone marrow; (Ctrl n=7, cKO n=6; mean ± SD). (D) Percentage of hematopoietic stem and progenitor cells in Bone marrow of control (Ctrl) or *Map4k4*-cKO (cKO) mice; LT-HSC, Long-term hematopoietic stem cells; ST-HSC, Short-term hematopoietic stem cells; MPP, Multipotent blood progenitors; CMP, Common Myeloid Progenitor; GMP, Granulocyte-Macrophage Progenitor; BM, bone marrow; (Ctrl n=7, cKO n=6; mean ± SD). (E) Numbers of hematopoietic stem and progenitor cells in Bone marrow of control (Ctrl) or *Map4k4*-cKO (cKO) mice; LT-HSC, Long-term hematopoietic stem cells; ST-HSC, Short-term hematopoietic stem cells; MPP, Multipotent blood progenitors; CMP, Common Myeloid Progenitor; GMP, Granulocyte-Macrophage Progenitor; BM, bone marrow; (Ctrl n=7, cKO n=6; mean ± SD). (F) The number of primitive erythroid progenitor cells (BFU-E) colonies formed by 25,000 whole bone marrow cells from control (Ctrl) or *Map4k4*-cKO (cKO) BM, bone marrow; (n=4; mean ± SD). (G) Numbers of Eos cells in the Bone marrow of control (Ctrl) or *Map4k4*-cKO (cKO) mice; BM, bone marrow; Eso, eosinophils; (Ctrl n=7, cKO n=6; mean ± SD). (H) Numbers of Mon cells in the Bone marrow of control (Ctrl) or *Map4k4*-cKO (cKO) mice; BM, bone marrow; Mon, monocyte (Ctrl n=7, cKO n=6; mean ± SD). (I) PB LYM of control (Ctrl) or *Map4k4*-cKO (cKO) mice; PB, peripheral blood; LYM, lymphocyte (Ctrl n=7, cKO n=6; mean ± SD). (J) PB RBC numbers, HGB, MCH, and MCV of control (Ctrl) or *Map4k4*-cKO (cKO) mice; PB, peripheral blood; RBC, Red blood cell; HGB, Hemoglobin; MCH, Mean corpuscular hemoglobin; MCV, Mean corpuscular volume; (Ctrl n=7, cKO n=6; mean ± SD). (K) PB PLT, PCT of control (Ctrl) or *Map4k4*-cKO (cKO) mice; PB, peripheral blood; MPLT, Platelets; PCT, Plateletcrit; (Ctrl n=7, cKO n=6; mean ± SD). (L) Numbers of DCs in the spleen of control (Ctrl) or *Map4k4*-cKO (cKO) mice; SP, Spleen; DCs, Dendritic cells; (Ctrl n=7, cKO n=6; mean ± SD). (M) Numbers of Mø in the spleen of control (Ctrl) or *Map4k4*-cKO (cKO) mice; SP, Spleen; Mø, Macrophage; (Ctrl n=7, cKO n=6; mean ± SD). Mann-Whitney U test.
(PDF)

**S4 Fig. Quality control and cell type identification of scRNA-seq.** (A-C) UMAP of the 14,725 cells profiled here, with each cell color-coded for (A) Control or cKO, (B) the number of counts per barcode, and (C) the number of detected genes per cell. (D) Dot plot of marker genes. The dot plot shows the average expression level (the intensity of green) and percentage of expressed cells (the dot size). (E) UMAP of the 14,725 cells colored by cell type. (F) Monocle trajectories of neutrophils colored by pseudotime. (G) Monocle trajectories of neutrophils colored by cell type. Each dot represents a single cell. Cell orders are inferred from the expression of the most variable genes across all cells. Trajectory directions were determined by biological prior. (H) ROS generation of neutrophils in the spleen (Left) and peripheral blood (Right) of control (Ctrl) or *Map4k4*-cKO (cKO) mice; MFI, mean fluorescent intensity; ROS, reactive oxygen species; (n=6; mean ± SD). Mann-Whitney U test. (I) The percentage of phagocytic neutrophils in the spleen (Left) and peripheral blood (Right) of control (Ctrl) or *Map4k4*-cKO (cKO) mice; (n=6; mean ± SD). Mann-Whitney U test.
(PDF)

**S5 Fig. hdWGCNA co-expression module network and gene set enrichment analysis (A-E) Hub gene networks for each Neu co-expression module.** The top 25 hub genes ranked by kME are visualized. Nodes represent genes, and edges represent co-expression links. (F) Gene ontology enrichment analyses of hub genes from co-expression module M2 M3 M4. The dot size represented the number of genes. The color scale represented the adjusted p value. (G)

Heatmap showing log2 (foldchange) in gene expression of the representative top300 apoptosis-related DEGs between control and cKO. The asterisks mean padjust < 0.05 in corresponding cells. (H) Percentages of annexin V$^+$ and SYTOX$^-$ cells in bone marrow neutrophil progenitor cells of control (Ctrl) or *Map4k4*-cKO (cKO) mice; MB, myeloblasts, MB; PM, promyelocytes, PM; MC, myelocytes; BM, bone marrow (n=10; mean ± SD). Mann-Whitney U test. (I) Representative FACS analysis of HL-60 cells treated with 0-20 μM MAP4K4 inhibitor (PF-06260933, PF) for staining of annexin V and SYTOX was shown (left). percentages of annexin V$^+$ and SYTOX$^-$ cells (right) (n=4; mean ± SD). Mann-Whitney U test. (J) Quantification of HL-60 cell numbers following treatment with a 10 μM concentration of the MAP4K4 inhibitor (PF-06260933, PF) was conducted over a period extending from 0 to 108 hours (n=4; mean ± SD). Mann-Whitney U test.
(PDF)

**S6 Fig.  Quality control and protein expression of mass spectrometry.** (A-B) The figure shows the distribution of peptide length and the corresponding m/z (mass-to-charge ratio) for peptides with different charge states, (A) for protein, and (B) for phosphorylation. Each point represents a peptide, with the color indicating its charge state. The histograms above and to the right of the scatter plot represent the distributions of peptide length and m/z, respectively, for peptides with different charge states. (C-D) The principal components analysis (PCA) results of 4 Ctrl samples (blue) and 4 MAP4K4 KO samples (red) for protein (C) and phosphorylation (D) from mass spectrometry. (E) GO-BP analysis of differential phosphorylation level proteins between Ctrl and MAP4K4 KO samples. Selected GO terms with Benjamini-Hochberg-corrected p values < 0.05 (one-sided Fisher's exact test) are shown. The dot size represented the number of genes. The color scale represented the adjusted p value. (F) Western blot (WB) for STAT5A and GAPDH in Ctrl and STAT5A overexpression cells (left), for STAT5B and GAPDH in Ctrl and STAT5B overexpression cells (right).
(PDF)

## Acknowledgments

We thank Prof. Chong Chen at Sichuan University for providing the MX1-Cre+ mouse. We thank Prof. Yuan Wang at Sichuan University for giving the Lentiviral vectors Plenti-CRISPR V2. We thank Jingjie PTM BioLab Co., Inc. (Hangzhou, China) for the analysis of Proteomics and Phosphoproteomics mass spectrometry. We thank Biomarker Technologies (Beijing, China) for scRNA-seq.

## Author contributions

**Conceptualization:** Guihua Wang, Dan Zhang, Zhifeng He, Bin Mao, Jing-wen Lin, Lu Chen.

**Data curation:** Guihua Wang, Dan Zhang, Zhifeng He.

**Formal analysis:** Guihua Wang, Dan Zhang, Zhifeng He, Jing-wen Lin, Lu Chen.

**Funding acquisition:** Chao Tang, Jing-wen Lin, Lu Chen.

**Investigation:** Guihua Wang, Dan Zhang, Zhifeng He.

**Methodology:** Guihua Wang, Dan Zhang, Zhifeng He, Bin Mao, Xiao Hu, Li Chen, Qingxin Yang, Zhen Zhou, Yating Zhang, Kepan Linghu, Zijie Xu, Jing-wen Lin, Lu Chen.

**Project administration:** Guihua Wang, Dan Zhang, Jing-wen Lin, Lu Chen.

**Resources:** Guihua Wang, Bin Mao, Xiao Hu, Li Chen, Qingxin Yang, Zhen Zhou, Yating Zhang, Kepan Linghu, Chao Tang, Jing-wen Lin, Lu Chen.

**Software:** Dan Zhang, Zhifeng He, Li Chen, Qingxin Yang, Kepan Linghu, Chao Tang, Zijie Xu, Defu Liu, Junwei Song, Huiying Wang, Yishan Lin, Ruihan Li, Lu Chen.

**Supervision:** Zhifeng He, Bin Mao, Jing-wen Lin, Lu Chen.

**Validation:** Guihua Wang, Dan Zhang, Zhifeng He.

**Visualization:** Guihua Wang, Dan Zhang, Zhifeng He.

**Writing – original draft:** Guihua Wang, Dan Zhang, Zhifeng He, Bin Mao, Xiao Hu, Li Chen, Qingxin Yang, Zhen Zhou, Yating Zhang, Zijie Xu, Defu Liu, Junwei Song, Huiying Wang, Yishan Lin, Jing-wen Lin, Lu Chen.

**Writing – review & editing:** Guihua Wang, Dan Zhang, Zhifeng He, Bin Mao, Xiao Hu, Li Chen, Qingxin Yang, Zhen Zhou, Yating Zhang, Zijie Xu, Defu Liu, Junwei Song, Huiying Wang, Jing-wen Lin, Lu Chen.

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
