## [Decision Letter · Decision Letter 0]

18 Dec 2024

PCOMPBIOL-D-24-01958

Machine learning-based prediction reveals kinase MAP4K4 regulates neutrophil differentiation through phosphorylating apoptosis-related proteins

PLOS Computational Biology

Dear Dr. Chen,

Thank you for submitting your manuscript to PLOS Computational Biology. After careful consideration, we feel that it has merit but does not fully meet PLOS Computational Biology's publication criteria as it currently stands. Therefore, we invite you to submit a revised version of the manuscript that addresses the points raised during the review process.

Please submit your revised manuscript within 60 days Feb 17 2025 11:59PM. If you will need more time than this to complete your revisions, please reply to this message or contact the journal office at ploscompbiol@plos.org. Please include the following items when submitting your revised manuscript:

We look forward to receiving your revised manuscript.

Kind regards,

Robert F. Murphy

Guest Editor

PLOS Computational Biology

Padmini Rangamani

Section Editor

PLOS Computational Biology

**Additional Editor Comments:**

Two reviews are in hand and given the concerns that were raised I am making a decision without waiting for a third review. Both reviewers appreciate your study’s combination of computational modeling and collection of experimental support for computational predictions. However, both raise a range of issues ranging from relatively minor to quite significant. I believe that the most important issue to address is comparison with existing gene regulatory predictive models (issue 8 of Reviewer 1). I encourage you to respond to each reviewer concern either by making appropriate additions to the manuscript, or by providing careful explanations of the reasons that you have chosen not to.

**Journal Requirements:**

At this stage, the following Authors/Authors require contributions: Guihua Wang, Dan Zhang, Zhifeng He, Bin Mao, Xiao Hu, Li Chen, Qingxin Yang, Zhen Zhou, Yating Zhang, Kepan Linghu, Chao Tang, Zijie Xu, Defu Liu, Junwei Song, Huiying Wang, Yishan Lin, Ruihan Li, and Jing-wen Lin. Please ensure that the full contributions of each author are acknowledged in the "Add/Edit/Remove Authors" section of our submission form.

Potential Copyright Issues:

- Figure 4; Please confirm whether you drew the images / clip-art within the figure panels by hand. If you did not draw the images, please provide (a) a link to the source of the images or icons and their license / terms of use; or (b) written permission from the copyright holder to publish the images or icons under our CC BY 4.0 license. Alternatively, you may replace the images with open source alternatives. See these open source resources you may use to replace images / clip-art:

- The following Figure contains a logo or branding: Figure 1; We are not permitted to publish this under our CC-BY 4.0 license, even with permission. We ask that you please remove or replace it.

5) Please ensure that the funders and grant numbers match between the Financial Disclosure field and the Funding Information tab in your submission form. Note that the funders must be provided in the same order in both places as well. State the initials, alongside each funding source, of each author to receive each grant. For example: "This work was supported by the National Institutes of Health (####### to AM; ###### to CJ) and the National Science Foundation (###### to AM)." State what role the funders took in the study. If the funders had no role in your study, please state: "The funders had no role in study design, data collection and analysis, decision to publish, or preparation of the manuscript.".

**Reviewers' comments:**

Reviewer's Responses to Questions

**Comments to the Authors:**

Reviewer #1: This study presents a machine learning pipeline, MRGI (Myeloid Regulatory Gene Identifier), based on random forests to predict key genes involved in myeloid cell differentiation. The authors integrated features such as gene expression dynamics, physiological characteristics, pathological relevance, and conservation to train the model. The study highlights the MAP4K4 gene as a regulator of neutrophil differentiation and validates its function through in vivo and in vitro experiments, demonstrating that MAP4K4 modulates apoptosis-related protein phosphorylation during differentiation. However, I have several concerns, which are described in detail below.

1. In section "Step 1: Data collection", while authors describe the sources of data used in this study, they do not list the sample size of data downloaded from each source. I recommend that the authors explicitly list the sample sizes for each dataset used in the study, including the number of samples for both bulk RNA-seq and single-cell RNA-seq data from mouse and human sources. A table summarizing sample sizes, their sources and any preprocessing steps would be particularly helpful.

2. The study integrates data from mouse and human species, including bulk RNA-seq and single-cell RNA-seq data, but it does not provide detailed methods for integrating cross-species data or addressing potential biases between bulk and single-cell data. Specifically, how were homologous genes aligned between mouse and human datasets, and what strategies were used to mitigate platform-specific biases?

3. while the authors mention the use of Monocle for pseudotime calculation, details such as parameter settings and criteria for trajectory inference are missing.

4. In this part “Then the naïv e Bayesian (NB) algorithm used the P set as positive and the U set as negative. The NB classifier was then applied to assign a probabilistic class label to each gene in the U set. The probabilistic labels of the spies were used to decide which genes were most likely to be negative.”, the authors describe using a naïve Bayesian (NB) classifier to predict labels for the unlabeled dataset (U set). However, several critical details are missing. What are the respective sizes of the P set and U set, and how were these datasets partitioned into training and testing sets for training the NB classifier? Additionally, when the authors state, "assign a probabilistic class label to each gene in the U set," does the U set refer to the testing subset or the entire unlabeled dataset? Authors should clarify these points.

5. The authors used Gaussian Mixture Models (GMM) scores to get ‘non-functional’ genes, but the paper does not provide sufficient details about the training process for the GMM. Specifically, what dataset was used to train the GMM, and how were the parameters determined? Additionally, were any validation steps performed to confirm the accuracy of the classification?

6. The paper provides a brief description of the OntoVAE training process but lacks critical details necessary for reproducibility. Specifically, the authors should clarify the size of the dataset used for training OntoVAE, the architecture of the encoder and decoder (e.g., the number of layers and the type of networks used), the number of training epochs, and how the loss function was defined and optimized. Providing these details would allow readers to better understand the model's configuration

7. While the authors performed feature importance analysis using the Gini coefficient, the paper does not discuss experiments involving the removal of features with low importance scores. Without such analysis, it is unclear whether these less important features contribute noise or negatively impact model performance. The authors should consider conducting feature ablation experiments to assess the effect of removing low-importance features on the model's predictive performance.

8. The study lacks comparisons with other gene regulatory prediction models, such as CellOracle or SCENIC. Benchmarking against these models would highlight MRGI's unique advantages.

9. The study uses HEK293T cells to validate the function of MAP4K4. However, HEK293T cells are not derived from the hematopoietic or myeloid lineage and exhibit significant differences in biological characteristics compared to myeloid cells or neutrophils. This discrepancy may limit the relevance of the findings to the biological processes under investigation and potentially reduce the credibility of the conclusions. The authors should consider relevant cell lines to provide a more biologically relevant validation model and strengthen the study’s conclusions.

10. When constructing the balanced dataset for training the random forest, the authors downsampled the negative samples to 400, while the positive samples totaled 411. It is unclear why 400 negative samples were chosen instead of 411 to match the exact size of the positive set. This choice may introduce a slight imbalance in the dataset, potentially affecting model training.

11. The study uses AUC as the sole evaluation metric for model performance. I recommend including additional evaluation metrics, such as PRC (precision-recall curve), F1-score, MCC, and ACC, to provide a more comprehensive assessment of the model’s performance.

12. The README file currently outlines only the basic framework of this work. It should include detailed instructions on how to use the method, provide a toy example, and offer a clear explanation of the tool's functionality, purpose, and applicable scenarios. Incorporating these elements would greatly enhance its clarity and usability. Additionally, the datasets used in this work should be made available on the GitHub repository to further improve accessibility and comprehensiveness.

13. There are several grammar mistakes that can easily be corrected with Grammarly etc, for example: “We conducted in vitro studies using HL-60 human leukemia cells, which can be induced to a neutrophil-like (CD11b+) state...”

14. In Fig. 2H and 2I, the capitalization of ligand labels is innconsistent. Please standardize all ligand labels to uppercase (e.g., "TIGAR").

Reviewer #2: The authors developed a machine learning pipeline called myeloid regulatory gene identifier (MRGI), which uses a random forest model to predict potential genes involved in myeloid cell differentiation. Here, a positive unlabeled (PU) learning algorithm was used to train a model where 411 genes belonging to the regulation of myeloid differentiation was used as a positive set. The model predictions not limited to TFs included other genes such as enzymes, membrane proteins, and RBPs. The model predicted three novel genes namely, MAP4K4, FIG4, and TIGAR to be involved in myeloid cell differentiation, of which MAP4K4 was shown to be important for neutrophils. The authors showed that loss of MAP4K4 resulted in neutropenia in mice and MAP4K4 regulated apoptosis in neutrophil differentiation by modulating phosphorylation of STAT5. While MAP4K4 is shown to be a promising target for neutrophil differentiation based on the computational analysis and the experimental validation, given the scope of the journal and its audience, the manuscript could be improved by addressing the following concerns:

1. The authors used four broad categories that comprised 41 features in total to train the model. Most of these features are specifically associated with neutrophils, granulocytes, and basophils - cells originating from GMP. In contrast, fewer to none are specifically associated to other myeloid cells such as dendritic cells or macrophages. This is more evident in the physiological characteristics category, the inclusion of leukocyte count may reduce specificity, as leukocytes also include lymphocytes. To improve the model's specificity, I suggest running the model using only neutrophil-associated features or focusing on specific myeloid cells. Furthermore, the authors use genes associated to “regulation of myeloid cell differentiation” comprising of 411 genes as the only positive gene set for the PU-learning algorithm. This can further reduce the specificity. I strongly suggest using features and positive gene sets that are specific to each myeloid cells or better focus only on neutrophils as this would improve sensitivity of the model and align with the later experimental validation.

2. The authors generate MAP4KA IFN-inducible KO, confirm delay in differentiation and analyse the rate of apoptosis in neutrophil subsets. Also differences in scRNA-seq data, between wt and cKO, appeared to be very nuanced - are the functional properties of MAP4KA KO affected as well? ROS, NET, phagocytosis etc?

Minor comments:

3. Line 87-91. All features or any of the four features? Elaborate.

4. Line 93-97. How are the datasets between mouse and human datasets integrated or compared?

5. Line 99-100. Elaborate further, are the dynamic and differential analysis done on all myeloid cells or specifically neutrophils and their progenitors? Please include details.

6. Line 113/Table S2. Would be useful to include the type of gene as a column in table S2, does the proportion of the type in the positive/negative set influence the type of gene as seen in Figure 2E. Essentially, is the MRGI score of enzymes significantly higher than other protein types because there are more enzymes in the training data?

7. Line 119. Kindly specify the type of GO analysis used here.

8. Line139. How are these peak ranges defined? Unclear.

9. Line 154. Fig2B. Would be interesting to see this figure for all features in Table S1. Do all features have significant trend between functional vs. non-functional.

10. Line 161. Kindly specify the type of GO analysis used here.

11. Line 171-177. Explanation insufficient, to improve readability, kindly elaborate further on OntoVAE, the datasets used and the accuracy over the predictions.

12. Fig3A – Scale is missing.

13. Fig 3B – At what time point was the inhibitor given?

14. Line 256. Elaborate on the batch correction method used.

15. Fig6D. Kindly detail other pathways that are differential between KO and WT.

**Have the authors made all data and (if applicable) computational code underlying the findings in their manuscript fully available?**

Reviewer #1: Yes

Reviewer #2: Yes

PLOS authors have the option to publish the peer review history of their article (what does this mean? ). If published, this will include your full peer review and any attached files.

**Do you want your identity to be public for this peer review?** For information about this choice, including consent withdrawal, please see our Privacy Policy .

Reviewer #1: No

Reviewer #2: No

**Figure resubmission:**
---

## [Decision Letter · Decision Letter 1]

6 Feb 2025

PCOMPBIOL-D-24-01958R1

Machine learning-based prediction reveals kinase MAP4K4 regulates neutrophil differentiation through phosphorylating apoptosis-related proteins

PLOS Computational Biology

Dear Dr. Chen,

Thank you for submitting your manuscript to PLOS Computational Biology. After careful consideration, we feel that it has merit but does not fully meet PLOS Computational Biology's publication criteria as it currently stands. Therefore, we invite you to submit a revised version of the manuscript that addresses the points raised during the review process.

Please submit your revised manuscript within 30 days Apr 08 2025 11:59PM. If you will need more time than this to complete your revisions, please reply to this message or contact the journal office at ploscompbiol@plos.org. Please include the following items when submitting your revised manuscript:

We look forward to receiving your revised manuscript.

Kind regards,

Robert F. Murphy

Guest Editor

PLOS Computational Biology

Padmini Rangamani

Section Editor

PLOS Computational Biology

**Additional Editor Comments :**

Both reviewers felt that you had addressed their major concerns but that some minor issues remain. Please address each of these in your resubmission.

**Reviewers' comments:**

Reviewer's Responses to Questions

Reviewer #1: The authors have done a thorough job addressing the previous comments. However, there is still one remaining concern. For full reproducibility, all the datasets used in this study should be made available in the GitHub repository, rather than providing only example datasets. Additionally, the notebook OntoVAE.ipynb cannot currently be executed because certain files (e.g., data/GO_symbol_ontobj.pickle) are missing from the repository. Please ensure that every file required by OntoVAE.ipynb is included in the repository. Finally, it would be highly beneficial to provide the trained model checkpoints used in the paper so that the results can be fully reproduced.

Reviewer #2: We thank the authors for making the necessary changes in the first round of revision, the manuscript has improved in clarity and reads well now. Following are some minor comments to improve further.

1. Please use consistent capitalization for gene names. For eg., Either MAP4K4 or Map4k4, currently there seems to be a mix of both.

2. Line 136: Please mention in text the total number of pathways that went into the positive gene set. If not already done, in supplementary please tabulate the neutrophil associated pathways and their associated genes. Also include genes taken from literature (with reference). This would be helpful for addition of newly identified neutrophil-associated genes to the positive gene set in the future.

3. Line 225 & 562 is unclear: Why was the in silco KO performed on 2569 genes? Isn’t it supposed be 4786, as reported in line 173-175? I hope this is a typo. The 2569 genes comprise only of enzymes (line 210). However, your resulting 12 genes (lines 257) comprise of other types of genes. Please explain/correct.

4. The third limitation in discussion (line 513-515) seems contradictory to the final statement (line 518-520). The authors can briefly propose methods to extend the approach to other lymphoid and erythroid differentiation states. Such as cell-specific features, positive gene-sets, etc.

5. The authors mention TF’s such as PU.1, CEBPA, and GFI-1 in their introduction. It would be intersesting to see their NeuRGI scores. Essentially, do the well know TF’s important for neutrophil maturation/function also have high scores? If so, this can serve as an independent validation of the method. Worth including in the discussion.

**Have the authors made all data and (if applicable) computational code underlying the findings in their manuscript fully available?**

Reviewer #1: None

Reviewer #2: Yes

PLOS authors have the option to publish the peer review history of their article (what does this mean? ). If published, this will include your full peer review and any attached files.

**Do you want your identity to be public for this peer review?** For information about this choice, including consent withdrawal, please see our Privacy Policy .

Reviewer #1: No

Reviewer #2: No

**Figure resubmission:**
---

## [Editor Report · Decision Letter 2]

14 Feb 2025

Dear Professor Chen,

We are pleased to inform you that your manuscript 'Machine learning-based prediction reveals kinase MAP4K4 regulates neutrophil differentiation through phosphorylating apoptosis-related proteins' has been provisionally accepted for publication in PLOS Computational Biology.

Best regards,

Robert F. Murphy

Guest Editor

PLOS Computational Biology

Padmini Rangamani

Section Editor

PLOS Computational Biology

---

## [Editor Report · Acceptance letter]

PCOMPBIOL-D-24-01958R2

Machine learning-based prediction reveals kinase MAP4K4 regulates neutrophil differentiation through phosphorylating apoptosis-related proteins

Dear Dr Chen,

I am pleased to inform you that your manuscript has been formally accepted for publication in PLOS Computational Biology. Your manuscript is now with our production department and you will be notified of the publication date in due course.

With kind regards,

Anita Estes
